# Refining the impact of genetic evidence on clinical success

Eric Vallabh Minikel[1], Jeffery L. Painter[2,5], Coco Chengliang Dong[3] & Matthew R. Nelson[3,4 ✉]

The cost of drug discovery and development is driven primarily by failure[1], with only about 10% of clinical programmes eventually receiving approval[2–4]. We previously estimated that human genetic evidence doubles the success rate from clinical development to approval[5]. In this study we leverage the growth in genetic evidence over the past decade to better understand the characteristics that distinguish clinical success and failure. We estimate the probability of success for drug mechanisms with genetic support is 2.6 times greater than those without. This relative success varies among therapy areas and development phases, and improves with increasing confidence in the causal gene, but is largely unaffected by genetic effect size, minor allele frequency or year of discovery. These results indicate we are far from reaching peak genetic insights to aid the discovery of targets for more effective drugs.

Human genetics is one of the only forms of scientific evidence that can demonstrate the causal role of genes in human disease. It provides a crucial tool for identifying and prioritizing potential drug targets, providing insights into the expected effect (or lack thereof[6]) of pharmacological engagement, dose–response relationships[7–10] and safety risks[6,11–13]. Nonetheless, many questions remain about the application of human genetics in drug discovery. Genome-wide association studies (GWASs) of common, complex traits, including many diseases, generally identify variants of small effect. This contributed to early scepticism of the value of GWASs[14]. Anecdotally, such variants can point to highly successful drug targets[7–9], and yet, genetic support from GWASs is somewhat less predictive of drug target advancement than support from Mendelian diseases[5,15].

In this paper we investigate several open questions regarding the use of genetic evidence for prioritizing drug discovery. We explore the characteristics of genetic associations that are more likely to differentiate successful from unsuccessful drug mechanisms, exploring how they differ across therapy areas and among discovery and development phases. We also investigate how close we may be to saturating the insights we can gain from genetic studies for drug discovery and how much of the genetically supported drug discovery space remains clinically unexplored.

To characterize the drug development pipeline, we filtered Citeline Pharmaprojects for monotherapy programmes added since 2000 annotated with a highest phase reached and assigned both a human gene target (usually the gene encoding the drug target protein) and an indication defined in Medical Subject Headings (MeSH) ontology. This resulted in 29,476 target–indication (T–I) pairs for analysis (Extended Data Fig. 1a). Multiple sources of human genetic associations totalled 81,939 unique gene–trait (G–T) pairs, with traits also mapped to MeSH terms. Intersection of these datasets yielded an overlap of 2,166 T–I and G–T pairs (7.3%) for which the indication and the trait MeSH terms had a similarity ≥0.8; we defined these T–I pairs as possessing genetic support (Extended Data Figs. 1b and 2a and Methods). The probability of having

genetic support, or P(G), was higher for launched T–I pairs than those in historical or active clinical development (Fig. 1a). In each phase, P(G) was higher than previously reported[5,15], owing, as expected[15,16], more to new G–T discoveries than to changes in drug pipeline composition (Extended Data Fig. 3a–f). For ensuing analyses, we considered both historical and active programmes. We defined success at each phase as a T–I pair transitioning to the next development phase (for example, from phase I to II), and we also considered overall success—advancing from phase I to a launched drug. We defined relative success (RS) as the ratio of the probability of success, P(S), with genetic support to the probability of success without genetic support (Methods). We tested the sensitivity of RS to various characteristics of genetic evidence. RS was sensitive to the indication–trait similarity threshold (Extended Data Fig. 2a), which we set to 0.8 for all analyses herein. RS was >2 for all sources of human genetic evidence examined (Fig. 1b). RS was highest for Online Mendelian Inheritance in Man (OMIM) (RS = 3.7), in agreement with previous reports[5,15]; this was not the result of a higher success rate for orphan drug programmes (Extended Data Fig. 2b), a designation commonly acquired for rare diseases. Rather, it may owe partly to the difference in confidence in causal gene assignment between Mendelian conditions and GWASs, supported by the observation that the RS for Open Targets Genetics (OTG) associations was sensitive to the confidence in variant-to-gene mapping as reflected in the minimum share of locus-to-gene (L2G) score (Fig. 1c). The differences common and rare disease programmes face in regulatory and reimbursement environments[4] and differing proportions of drug modalities[9] probably contribute as well. OMIM and GWAS support were synergistic with one another (Supplementary Fig. 2b). Somatic evidence from IntOGen had an RS of 2.3 in oncology (Extended Data Fig. 2c), similar to GWASs, but analyses below are limited to germline genetic evidence unless otherwise noted.

As sample sizes grow ever larger with a corresponding increase in the number of unique G–T associations, some expect[17] the value of GWAS genetic findings to become less useful for the purpose of drug

[1]Stanley Center for Psychiatric Research, Broad Institute, Cambridge, MA, USA. [2]JiveCast, Raleigh, NC, USA. [3]Deerfield Management Company LP, New York, NY, USA. [4]Genscience LLC, New York, NY, USA. [5]Present address: GlaxoSmithKline, Research Triangle Park, NC, USA. ✉e-mail: mnelson@genscience.com

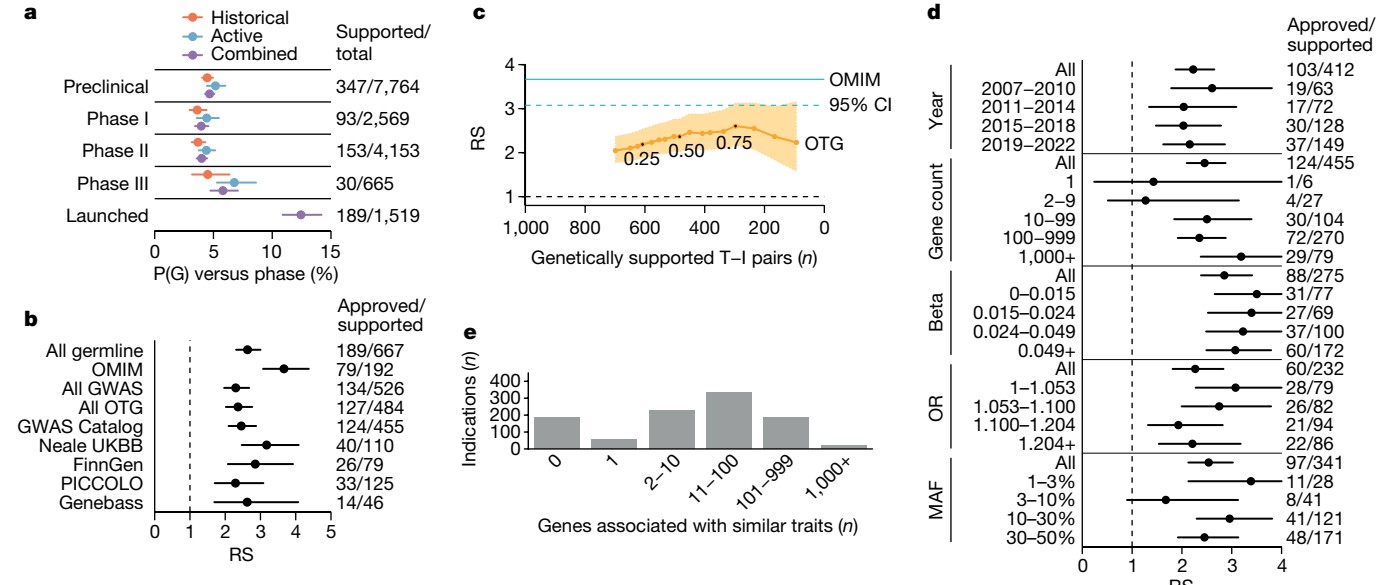

**Fig. 1 | Impact of genetic evidence characteristics on RS. a**, Proportion of T–I pairs with genetic support, P(G), as a function of highest phase reached. *n* at right: denominator, number of T–I pairs per phase; numerator, number that are genetically supported. **b**, Sensitivity of phase I–launch RS to source of human genetic association. GWAS Catalog, Neale UKBB and FinnGen are subsets of OTG. *n* at right: denominator, number of T–I pairs with genetic support from each source; numerator, number of those launched. Note that RS is calculated from a 2 × 2 contingency table (Methods). Total *n* = 13,022 T–I pairs. **c**, Sensitivity of RS to L2G share threshold among OTG associations. Minimum L2G share threshold is varied from 0.1 to 1.0 in increments of 0.05 (labels); RS (*y* axis) is plotted against the number of clinical (phase I+) programmes with genetic support from OTG (*x* axis). **d**, Sensitivity of RS for OTG GWAS-supported T–I

pairs to binned variables: (1) year that T–I pair first acquired human genetic support from GWASs, excluding replications and excluding T–I pairs otherwise supported by OMIM; (2) number of genes exhibiting genetic association to the same trait; (3) quartile of effect size (beta) for quantitative traits; (4) quartile of effect size (odds ratio, OR) for case/control traits standardized to be >1 (that is, 1/OR if <1); (5) order of magnitude of minor allele frequency bins. *n* at right as in **b**. Total *n* = 13,022 T–I pairs. **e**, Count of indications ever developed in Pharmaprojects (*y* axis) by the number of genes associated with traits similar to those indications (*x* axis). Throughout, error bars or shaded areas represent 95% CIs (Wilson for P(G) and Katz for RS) whereas centres represent point estimates. See Supplementary Fig. 1 for the same analyses restricted to drugs with a single known target.

target selection. We explored this in several ways. We investigated the year that genetic support for a T–I pair was first discovered, under the expectation that more common and larger effects are discovered earlier. Although there was a slightly higher RS for discoveries from 2007–2010 that was largely driven by early lipid and cardiovascular-related associations, the effect of year was overall non-significant (*P* = 0.46; Fig. 1d). Results were similar when replicate associations or OMIM discoveries were included (Extended Data Fig. 2d–f). We next divided up GWAS-supported drug programmes by the number of unique traits associated to each gene. RS nominally increased with the number of associated genes, by 0.048 per gene (*P* = 0.024; Fig. 1d). The reason is probably not that successful genetically supported programmes inspire other programmes, because most genetic support was discovered retrospectively (Extended Data Fig. 2g); the few examples of drug programmes prospectively motivated by genetic evidence were primarily for Mendelian diseases[9]. There were no statistically significant associations with estimated effect sizes (*P* = 0.90 and 0.57, for quantitative and binary traits, respectively; Fig. 1d and Extended Data Fig. 2h) or minor allele frequency (*P* = 0.26; Fig. 1d). That ever larger GWASs can continue to uncover support for successful targets is also illustrated by two recent large GWASs in type 2 diabetes (T2D)[18,19] (Extended Data Fig. 4).

Previously[5], we observed significant heterogeneity among therapy areas in the fraction of approved drug mechanisms with genetic support, but did not investigate the impact on probability of success[5]. Here, our estimates of RS from phase I to launch showed significant heterogeneity (*P* < 1.0 × 10⁻¹⁵), with nearly all therapy areas having estimates greater than 1; 11 of 17 were >2, and haematology, metabolic, respiratory and endocrine >3 (Fig. 2a–e). In most therapy areas, the impact of genetic evidence was most pronounced in phases II and III and

least impactful in phase I, corresponding to capacity to demonstrate clinical efficacy in later development phases. Accordingly, therapy areas differed in P(G) and in whether P(G) increased throughout clinical development or only at launch (Extended Data Fig. 5); data source and other properties of genetic evidence including year of discovery and effect size also differed (Extended Data Fig. 6). We also found that genetic evidence differentiated likelihood to progress from preclinical to clinical development for metabolic diseases (RS = 1.38; 95% confidence interval (95% CI), 1.25 to 1.54), which may reflect preclinical models that are more predictive of clinical outcomes. P(G) by therapy area was correlated with P(S) (*ρ* = 0.59, *P* = 0.013) and with RS (*ρ* = 0.72, *P* = 0.0011; Extended Data Fig. 7), which led us to explore how the sheer quantity of genetic evidence available within therapy areas (Fig. 2f and Extended Data Fig. 8a) may influence this. We found that therapy areas with more possible gene–indication (G–I) pairs supported by genetic evidence had significantly higher RS (*ρ* = 0.71, *P* = 0.0010; Fig. 2g), although respiratory and endocrine were notable outliers with high RS despite fewer associations.

We hypothesized that genetic support might be most pronounced for drug mechanisms with disease-modifying effects, as opposed to those that manage symptoms, and that the proportions of such drugs differ by therapy area[20,21]. We were unable to find data with these descriptions available for a sufficient number of drug mechanisms to analyse, but we reasoned that targets of disease-modifying drugs are more likely to be specific to a disease, whereas targets of symptom-managing drugs are more likely to be applied across many indications. We therefore examined the number and diversity of all-time launched indications per target. Launched T–I pairs are heavily skewed towards a few targets (Fig. 2h). Of 450 launched targets, the 42 with ≥10 launched indications comprise 713 (39%) of 1,806 launched

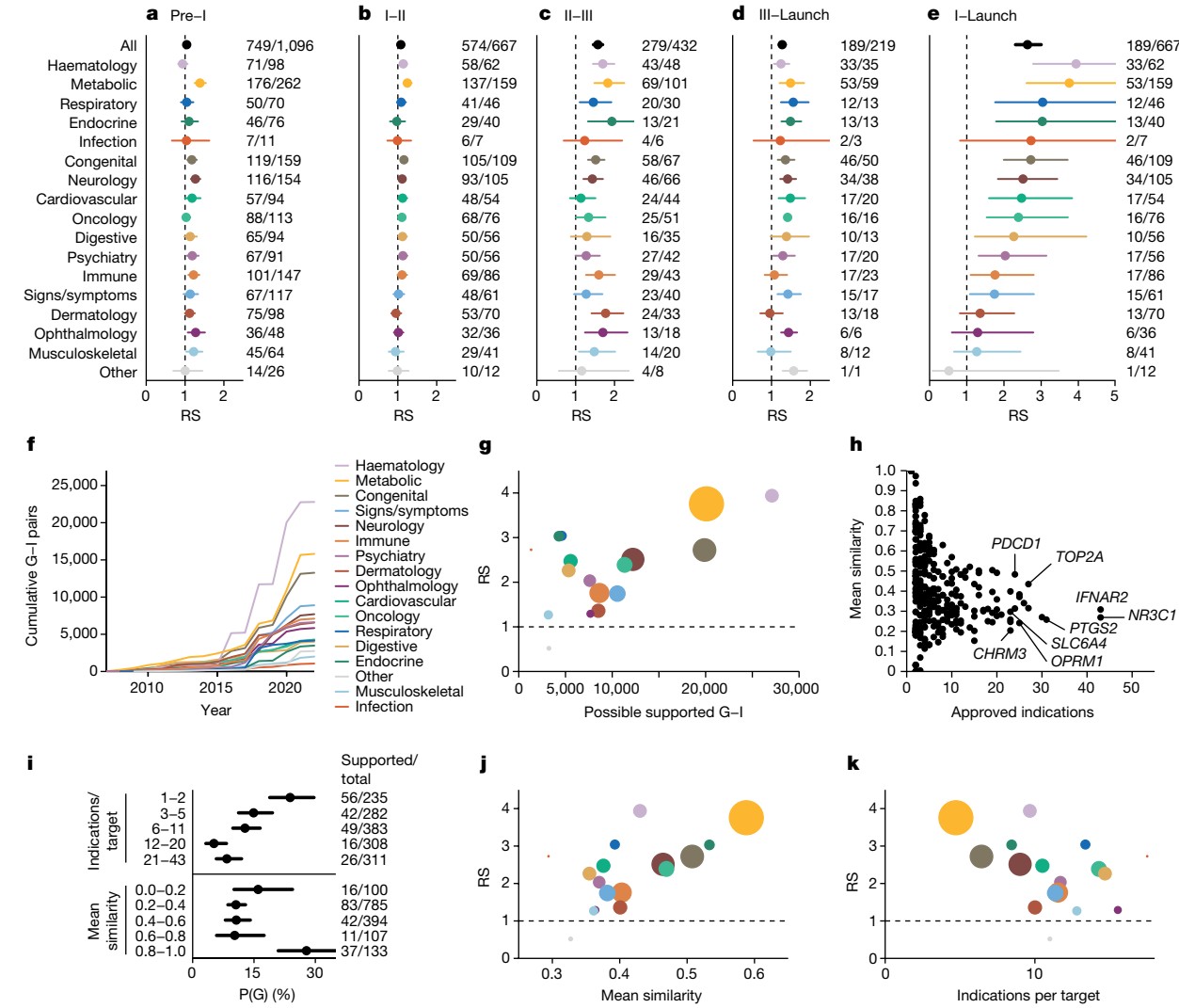

**Fig. 2 | Differences in RS between therapy areas and the number and diversity of indications per target. a–e**, RS by therapy area and phase transitions: preclinical to phase I (**a**), phase I to II (**b**), phase II to III (**c**), phase III to launch (**d**) and phase I to launch (**e**). *n* at right: denominator, T–I pairs with genetic support; numerator, number of those that succeeded in the phase transition indicated at the top of the panel. For 'all', total *n* = 22,638 preclinical, 13,022 reaching at least phase I, 7,223 reaching at least phase II and 2,184 reaching at least phase III. Total *n* for each therapy area is provided in Supplementary Table 27. **f**, Cumulative number of possible genetically supported G–I pairs in each therapy (*y* axis) as genetic discoveries have accrued over time (*x* axis). **g**, RS (*y* axis) by number of possible supported G–I pairs (*x* axis) across therapy areas, with dots coloured as in panels **a**–**e** and sized according to number of genetically supported T–I pairs in at least phase I.

**h**, Number of launched indications versus similarity of those indications, by approved drug target. **i**, Proportion of launched T–I pairs with genetic support, P(G), binned by quintile of the number of launched indications per target (top panel) or by mean similarity among launched indications (bottom panel). Targets with exactly 1 launched indication (6.2% of launched T–I pairs) are considered to have mean similarity of 1.0. *n* at right: denominator, total number of launched T–I pairs in each bin; numerator, number of those with genetic support. **j**, RS (*y* axis) versus mean similarity among launched indications per target (*x* axis) by therapy area. **k**, RS (*y* axis) versus mean count of launched indications per target (*x* axis). Throughout, error bars or shaded areas represent 95% CIs (Wilson for P(G) and Katz for RS) whereas centres represent point estimates. See Supplementary Fig. 2 for the same analyses restricted to drugs with a single known target.

T–I pairs (Fig. 2h). Many of these are used across diverse indications for management of symptoms such as inflammatory and immune responses (*NR3C1*, *IFNAR2*), pain (*PTGS2*, *OPRM1*), mood (*SLC6A4*) or parasympathetic response (*CHRM3*). The count of launched indications was inversely correlated with the mean similarity of those indications ($\rho = -0.72$, $P = 4.4 \times 10^{-84}$; Fig. 2h). Among T–I pairs, the probability of having genetic support increased as the number of launched indications decreased ($P = 6.3 \times 10^{-7}$) and as the similarity of a target's launched indications increased ($P = 1.8 \times 10^{-5}$; Fig. 2i). We observed a corresponding impact on RS, increasing in therapy areas for which the similarity among launched indications increased, and decreasing with increasing indications per target ($\rho = 0.74$, $P = 0.0010$, and $\rho = -0.62$, $P = 0.0080$, respectively; Fig. 2j,k).

Only 4.8% (284 of 5,968) of T–I pairs active in phases I–III possess human germline genetic support (Fig. 1a), similar to T–I pairs no longer in development (4.2%, 560 of 13,355), a difference that was not statistically significant ($P = 0.080$). We estimated (Methods) that only 1.1% of all genetically supported G–I relationships have been explored clinically (Fig. 3a), or 2.1% when restricting to the most similar indication. Given that the vast majority of proteins are classically 'undruggable', we explored the proportion of genetically supported G–I pairs that had been developed to at least phase I, as a function of therapy area across several classes of tractability and relevant protein families[22] (Fig. 3a). Within therapy areas, oncology kinases with germline evidence were the most saturated: 109 of 250 (44%) of all genetically supported G–I pairs had reached at least phase I; GPCRs for psychiatric indications

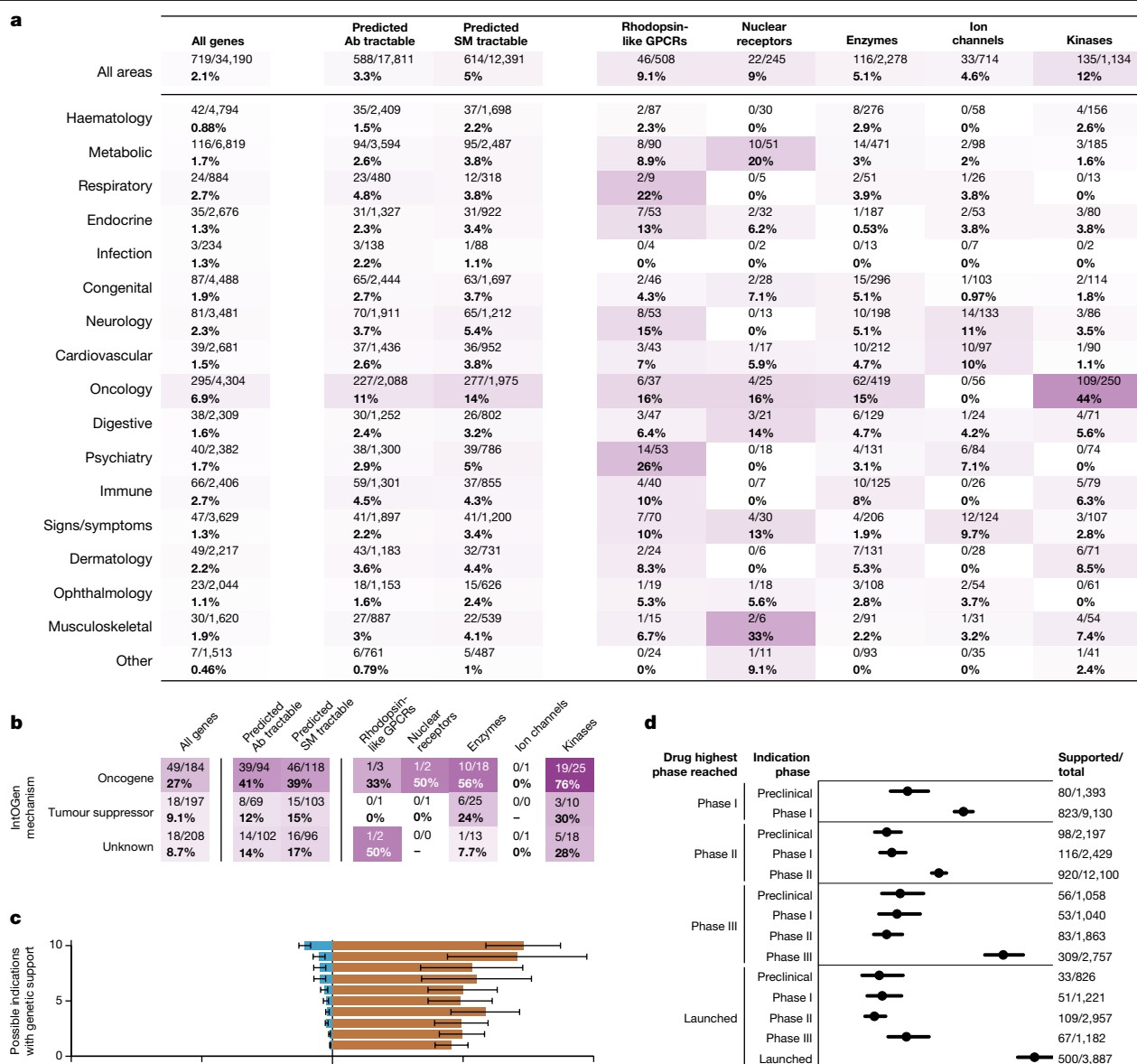

**Fig. 3 | Clinical investigation of drug mechanisms with genetic evidence.**
**a**, Heatmap of proportion of genetically supported T–I pairs that have been developed to at least phase I, by therapy area (*y* axis) and gene list (*x* axis). **b**, As panel **a**, but for genetic support from IntOGen rather than germline sources and grouped by the direction of effect of the gene according to IntOGen (*y* axis), and also grouped by target rather than T–I pair. Thus, the denominator for each cell is the number of targets with at least one genetically supported indication, and each target counts towards the numerator if at least one genetically supported indication has reached phase I. **c**, Of targets that have reached phase I for any indication, and have at least one genetically supported indication, the mean count (*x* axis) of genetically supported (left) and unsupported (right) were also notable (14 of 53, 26%). Grouping by target rather than G–I pair, 3.6% of genetically supported targets have been pursued for any genetically supported indication (Extended Data Fig. 8). Of possible genetically supported G–I pairs, most (68%) arose from OTG associations, mostly in the past 5 years (Fig. 2f). Such low use is partly due to recent emergence of most genetic evidence (Extended Data Figs. 2f,g and 7a), as drug programmes prospectively supported by human genetics have had a mean lag time from genetic association of 13 years to first trial[21] and 21 years to approval[9]. Because some types of targets may be indications pursued, binned by the number of possible genetically supported indications (*y* axis). The centre is the mean and bars are Wilson 95% CIs. *n* = 1,147 targets. **d**, Proportion of D–I pairs with genetic support, P(G) (*x* axis), as a function of each D–I pair's phase reached (inner *y*-axis grouping) and the drug's highest phase reached for any indication (outer *y*-axis grouping). The centre is the exact proportion and bars are Wilson 95% CIs. The *n* is indicated at the right, for which the denominator is the total number of D–I pairs in each bin, and the numerator is the number of those that are genetically supported. See Supplementary Fig. 3 for the same analyses restricted to drugs with a single known target. Ab, antibody; SM, small molecule.

more readily tractable by antagonists than agonists, we also grouped by target and examined human genetic evidence by direction of effect for tumour suppressors versus oncogenes (Fig. 3b), identifying a few substrata for which a majority of genetically supported targets had been pursued to at least phase I for at least one genetically supported indication. Oncogene kinases received the most attention, with 19 of 25 (76%) reaching phase I.

To focus on demonstrably druggable proteins, we further restricted the analysis to targets with both (1) any programme reaching phase

I, and (2) ≥1 genetically supported indications. Of 1,147 qualifying targets, only 373 (33%) had been pursued for one or more supported indications (Fig. 3c), and most (307, 27%) of these targets were pursued for indications both with and without genetic support. Overall, an overwhelming majority of development effort has been for unsupported indications, at a 17:1 ratio. Within this subset of targets, we asked whether genetic support was predictive of which indications would advance the furthest. Grouping active and historical programmes by drug–indication (D–I) pair, we found that the odds of advancing to a later stage in the pipeline are 82% higher for indications with genetic support ($P = 8.6 \times 10^{-73}$; Fig. 3d).

Although there has been anecdotal support—such as the *HMGCR* example—to argue that genetic effect size may not matter in prioritizing drug targets, here we provide systematic evidence that small effect size, recent year of discovery, increasing number of genes identified or higher associated allele frequency do not diminish the value of GWAS evidence to differentiate clinical success rates. One reason for this is probably because genetic effect size on a phenotype rarely accounts for the magnitude of genetic effect on gene expression, protein function or some other molecular intermediate. In some circumstances, genetic effect sizes can yield insights into anticipated drug effects. This is best illustrated for cardiovascular disease therapies, for which genetic effects on cholesterol and disease risk and treatment outcomes are correlated[23]. A limitation is that, other than Genebass, we did not include whole exome or whole genome sequencing association studies, which may be more likely to pinpoint causal variants. Moreover, all of our analyses are naive to direction of genetic effect (gain versus loss of gene function) as this is unknown or unannotated in most datasets used here.

Our results argue for continuing investment to expand GWAS-like evidence, particularly for many complex diseases with treatment options that fail to modify disease. Although genetic evidence has value across most therapy areas, its benefit is more pronounced in some areas than others. Furthermore, it is possible that the therapy areas for which genetic evidence had a lower impact have seen more focus on symptom management. If so, we would predict that for drugs aimed at disease modification, human genetics should ultimately prove highly valuable across therapy areas.

The focus of this work has been on the RS of drug programmes with and without genetic evidence, limited to drug mechanisms that have entered clinical development. This metric does not address the probability that a gene associated with a disease, if targeted, will yield a successful drug. At the early stage of target selection, is evidence of a large loss-of-function effect in one gene usually a better choice than a small non-coding single nucleotide polymorphism (SNP) effect on the same phenotype in another? We explored this question for T2D studies referenced above. When these GWASs quadrupled the number of T2D-associated genes from 217 to 862, new genetic support was identified for 7 of 95 mechanisms in clinical development whereas the number supported increased from 5 to 7 of 12 launched drug mechanisms. Thus, RS has remained high in light of new GWAS data. One can also, however, consider the proportion of genetic associations that are successful drug targets. Of the 7 targets of launched drugs with genetic evidence, 4 had Mendelian evidence (in addition to pre-2020 GWAS evidence), out of a total of 19 Mendelian genes related to T2D (21%). One launched T2D target had only GWAS (and no Mendelian) evidence among 217 GWAS-associated genes before 2020 (0.46%), whereas 2 launched targets were among 645 new GWAS associations since 2020 (0.31%). At least in this example, the 'yield' of genetic evidence for successful drug mechanisms was greatest for genes with Mendelian effects, but similar between earlier and later GWASs. Clearly, just because genetic associations differentiate clinical stage drug targets from launched ones, does not mean that a large fraction of associations will be fruitful. Moreover, genetically supported targets may be more likely to require upregulation, to be druggable only by more challenging modalities[4,9] or to enjoy narrower use across indications. More work is required to better understand the challenges of target identification and prioritization given the genetic evidence precondition.

The utility of human genetic evidence in drug discovery has had firm theoretical and empirical footing for several years[5,7,15]. If the benefit of this evidence were cancelled out by competitive crowding[24], then currently active clinical phases should have higher rates of genetic support than their corresponding historical phases, and might look similar to, or even higher than, launched pairs. Instead, we find that active programmes possess genetic support only slightly more often than historical programmes and remain less enriched for genetic support than launched drugs. Meanwhile, only a tiny fraction of classically druggable genetically supported G–I pairs have been pursued even among targets with clinical development reported. Human genetics thus represents a growing opportunity for novel target selection and improving indication selection for existing drugs and drug candidates. Increasing emphasis on drug mechanisms with supporting genetic evidence is expected to increase success rates and lower the cost of drug discovery and development.

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

## Methods

### Definition of metrics

Except where otherwise noted, we define genetic support of a drug mechanism (that is, a T–I pair) as a genetic association mapped to the corresponding target gene for a trait that is ≥0.8 similar to the indication (see MeSH term similarity below). We defined P(G) as the proportion of drug mechanisms satisfying the above definition of genetic support. P(S) is the proportion of programmes in one phase that advance to a subsequent phase (for instance, phase I to phase II). Overall P(S) from phase I to launched is the product of P(S) at each individual phase. RS is the ratio of P(S) for programmes with genetic support to P(S) for programmes lacking genetic support, which is equivalent to a relative risk or risk ratio. Thus, if $N$ denotes the total number of programmes that have reached the reference phase, and $X$ denotes the number of those that advance to a later phase of interest, and the subscripts G and !G indicate the presence or absence of genetic support, then $P(G) = N_G/(N_G + N_{!G})$; $P(S) = (X_G + X_{!G})/(N_G + N_{!G})$; $RS = (X_G/N_G)/(X_{!G}/N_{!G})$. RS from phase I to launched is the product of RS at each individual phase. The count of 'programs' for $X$ and $N$ is T–I pairs throughout, except for Fig. 3d, which uses D–I pairs to specifically interrogate P(G) for which the same drug has been developed for different indications. For clarity, we note that whereas other recent studies[22,25] have examined the fold enrichment and overlap between genes with a human genetic support and genes encoding a drug target, without regard to similarity, herein all of our analyses are conditioned on the similarity between the drug's indication and the genetically associated trait.

### Drug development pipeline

Citeline Pharmaprojects[26] is a curated database of drug development programmes including preclinical, all clinical phases and launched (approved and marketed) drugs. It was queried via API (22 December 2022) to obtain information on drugs, targets, indications, phases reached and current development status. T–I pair was the unit of analysis throughout, except where otherwise indicated in the text (D–I pairs were examined in Fig. 3d). Current development status was defined as 'active' if the T–I pair had at least one drug still in active development, and 'historical' if development of all drugs for the T–I pair had ceased. Targets were defined as genes; as most drugs do not directly target DNA, this usually refers to the gene encoding the protein target that is bound or modulated by the drug. We removed combination therapies, diagnostic indication and programmes with no human target or no indication assigned. For most analyses, only programmes added to the database since 2000 were included, whereas for the count and similarity of launched indications per target, we used all launches for all time. Indications were considered to possess 'genetic insight'—meaning the human genetics of this trait or similar traits have been successfully studied—if they had ≥0.8 similarity to (1) an OMIM or IntOGen disease, or (2) a GWAS trait with at least 3 independently associated loci, on the basis of lead SNP positions rounded to the nearest 1 megabase. For calculating RS, we used the number of T–I pairs with genetic insight as the denominator. The rationale for this choice is to focus on indications for which there exists the opportunity for human genetic evidence, consistent with the filter applied previously[5]. However, we observe that our findings are not especially sensitive to the presence of this filter, with RS decreasing by just 0.17 when the filter is removed (Extended Data Fig. 3g,h). Note that the criteria for determining genetic insight are distinct from, and much looser than, the criteria for mapping GWAS hits to genes (see L2G scores under OTG below). Many drugs had more than one target assigned, in which case all targets were retained for T–I pair analyses. As a sensitivity test, running our analyses restricted to only drugs with exactly one target assigned yielded very similar results (Supplementary Figures).

### OMIM

OMIM is a curated database of Mendelian gene–disease associations. The OMIM Gene Map (downloaded 21 September 2023) contained 8,671 unique gene–phenotype links. We restricted to entries with phenotype mapping code 3 ('the molecular basis for the disorder is known; a mutation has been found in the gene'), removed phenotypes with no MIM number or no gene symbol assigned, and removed duplicate combinations of gene MIM and phenotype MIM. We used regular expression matching to further filter out phenotypes containing the terms 'somatic', 'susceptibility' or 'response' (drug response associations) and those flagged as questionable ('?'), or representing non-disease phenotypes ('['). A set of OMIM phenotypes are flagged as denoting susceptibility rather than causation ('{'); this category includes low-penetrance or high allele frequency association assertions that we wished to exclude, but also germline heterozygous loss-of-function mutations in tumour suppressor genes, for which the underlying mechanism of disease initiation is loss of heterozygosity, which we wished to include. We therefore also filtered out phenotypes containing '{' except for those that did contain the terms 'cancer', 'neoplasm', 'tumor' or 'malignant' and did not contain the term 'somatic'. Remaining entries present in OMIM as of 2021 were further evaluated for validity by two curators, and gene–disease combinations for which a disease association was deemed not to have been established were excluded from all analyses. All of the above filters left 5,670 unique G–T links. MeSH terms for OMIM phenotypes were then mapped using the EFO OWL database using an approach previously described[27], with further mappings from Orphanet, full text matches to the full MeSH vocabulary and, finally, manual curation, for a cumulative mapping rate of 93% (5,297 of 5,670). Because sometimes distinct phenotype MIM numbers mapped to the same MeSH term, this yielded 4,510 unique gene–MeSH links.

### OTG

OTG is a database of GWAS hits from published studies and biobanks. OTG version 8 (12 October 2022) variant-to-disease, L2G, variant index and study index data were downloaded from EBI. Traits with multiple EFO IDs were excluded as these generally represent conditional, epistasis or other complex phenotypes that would lack mappings in the MeSH vocabulary. Of the top 100 traits with the greatest number of genes mapped, we excluded 76 as having no clear disease relevance (for example, 'red cell distribution width') or no obvious marginal value (for example, excluded 'trunk predicted mass' because 'body mass index' was already included). Remaining traits were mapped to MeSH using the EFO OWL database, full text queries to the MeSH API, mappings already manually curated in PICCOLO (see below) or new manual curation. In total, 25,124 of 49,599 unique traits (51%) were successfully mapped to a MeSH ID. We included associations with $P < 5 \times 10^{-8}$. OTG L2G scores used for gene mapping are based on a machine learning model trained on gold standard causal genes[28]; inputs to that model include distance, functional annotations, expression quantitative trait loci (eQTLs) and chromatin interactions. Note that we do not use Mendelian randomization[29] to map causal genes, and even gene mappings with high L2G scores are necessarily imperfect. OTG provides an L2G score for the triplet of each study or trait with each hit and each possible causal gene. We defined L2G share as the proportion of the total L2G score assigned each gene among all potentially causal genes for that trait–hit combination. In sensitivity analyses we considered L2G share thresholds from 10% to 100% (Fig. 1b and Extended Data Fig. 3a), but main analyses used only genes with ≥50% L2G share (which are also the top-ranked genes for their respective associations). OTG links were parsed to determine the source of each OTG data point: the EBI GWAS catalog[30] ($n = 136,503$ hits with L2G share ≥0.5), Neale UK Biobank (http://www.nealelab.is/uk-biobank; $n = 19,139$), FinnGen R6 (ref. 31) ($n = 2,338$) or SAIGE ($n = 1,229$).

## PICCOLO

PICCOLO[32] is a database of GWAS hits with gene mapping based on tests for colocalization without full summary statistics by using Probabilistic Identification of Causal SNPs (PICS) and a reference dataset of SNP linkage disequilibrium values. As described[32], gene mapping uses quantitative trait locus (QTL) data from GTEx ($n = 7,162$) and a variety of other published sources ($n = 6,552$). We included hits with GWAS $P < 5 \times 10^{-8}$, and with eQTL $P < 1 \times 10^{-5}$, and posterior probability H4 ≥ 0.9, as these thresholds were determined empirically[32] to strongly predict colocalization results.

## Genebass

Genebass[33] is a database of genetic associations based on exome sequencing. Genebass data from 394,841 UK Biobank participants (the '500K' release) were queried using Hail (19 October 2023). We used hits from four models: pLoF (predicted loss-of-function) or missense|LC (missense and low confidence LoF), each with sequencing kernel association test (SKAT) or burden tests, filtering for $P < 1 \times 10^{-5}$. Because the traits in Genebass are from UK Biobank, which is included in OTG, we used the OTG MeSH mappings established above.

## IntOGen

IntOGen is a database of enrichments of somatic genetic mutations within cancer types. We used the driver genes and cohort information tables (31 May 2023). IntOGen assigns each gene a mechanism in each tumour type; occasionally, a gene will be classified as a tumour suppressor in one type and an oncogene in another. We grouped by gene and assigned each gene its modal classification across cancers. MeSH mappings were curated manually.

## MeSH term similarity

MeSH terms in either Pharmaprojects or the genetic associations datasets that were Supplementary Concept Records (IDs beginning in 'C') were mapped to their respective preferred main headings (IDs beginning in 'D'). A matrix of all possible combinations of drug indication MeSH IDs and genetic association MeSH IDs was constructed. MeSH term Lin and Resnik similarities were computed for each pair as described[34,35]. Similarities of −1, indicating infinite distance between two concepts, were assigned as 0. The two scores were regressed against each other across all term pairs, and the Resnik scores were adjusted by a multiplier such that both scores had a range from 0 to 1 and their regression had a slope of 1. The two scores were then averaged to obtain a combined similarity score. Similarity scores were successfully calculated for 1,006 of 1,013 (99.3%) unique MeSH terms for Pharmaprojects indications, corresponding to 99.67% of Pharmaprojects T–I pairs, and for 2,260 of 2,262 (99.9%) unique MeSH terms for genetic associations, corresponding to >99.9% of associations.

## Therapeutic areas

MeSH terms for Pharmaprojects indications were mapped onto 16 top-level headings under the Diseases [C] and Psychiatry and Psychology [F] branches of the MeSH tree (https://meshb.nlm.nih.gov/treeView), plus an 'other'. The signs/symptoms area corresponds to C23 Pathological Conditions, Signs and Symptoms and contains entries such as inflammation and pain. Many MeSH terms map to >1 tree positions; these multiples were retained and counted towards each therapy area, except for the following conditions: for terms mapped to oncology, we deleted their mappings to all other areas; and 'other' was used only for terms that mapped to no other areas.

## Analysis of T2D GWASs

We included 19 genes from OMIM linked to Mendelian forms of diabetes or syndromes with diabetic features. For Vujkovic et al.[18], we considered as novel any genes with a novel nearest gene, novel coding variant or

a novel lead SNP colocalized with an eQTL with H4 ≥ 0.9. Non-novel nearest genes, coding variants and colocalized lead SNPs were considered established variants. For Suzuki et al.[19], we used the available L2G scores that OTG had assigned for the same lead SNPs in previously reported GWASs for other phenotypes, yielding mapped genes with L2G share >0.5 for 27% of loci. Genes were considered novel if absent from the Vujkovic analysis. Together, these approaches identified 217 established GWAS genes and 645 novel ones (469 from Vujkovic and 176 from Suzuki). We identified 347 unique drug targets in Pharmaprojects reported with a T2D or diabetes mellitus indication, including 25 approved. We reviewed the list of approved drugs and eliminated those for which there were questions around the relevance of the drug or target to T2D (*AKR1B1*, *AR*, *DRD1*, *HMGCR*, *IGF1R*, *LPL*, *SLC5A1*). Because Pharmaprojects ordinarily specifies the receptor as target for protein or peptide replacement therapies, we also remapped the minority of programmes for which the ligand, rather than receptor, had been listed as target (changing *INS* to *INSR*, *GCG* to *GCGR*). To assess the proportion of programmes with genetic support, we first grouped by drug and selected just one target, preferring the target with the earliest genetic support (OMIM, then established GWASs, then novel GWASs, then none). Next we grouped by target and selected its highest phase reached. Finally, we grouped by highest phase reached and counted the number of unique targets.

## Universe of possible genetically supported G–I pairs

In all of our analyses, targets are defined as human gene symbols, but we use the term G–I pair to refer to possible genes that one might attempt to target with a drug, and T–I pair to refer to genes that are the targets of actual drug candidates in development. To enumerate the space of possible G–I pairs, we multiplied the $n = 769$ Pharmaprojects indications considered here by the 'universe' of $n = 19,338$ protein-coding genes, yielding a space of $n = 14,870,922$ possible G–I pairs. Of these, $n = 101,954$ (0.69%) qualify as having genetic support per our criteria. A total of 16,808 T–I pairs have reached at least phase I in an active or historical programme, of which 1,155 (6.9%) are genetically supported. This represents an enrichment compared with random chance (OR = 11.0, $P < 1.0 \times 10^{-15}$, Fisher's exact test), but in absolute terms, only 1.1% of genetically supported G–I pairs have been pursued. A genetically supported G–I pair may be less likely to attract drug development interest if the indication already has many other potential targets, and/or if the indication is but the second-most similar to the gene's associated trait. Removing associations with many GWAS hits and restricting to the single most similar indication left a space of 34,190 possible genetically supported G–I pairs, 719 (2.1%) of which had been pursued. This small percentage might yet be perceived to reflect competitive saturation, if the vast majority of indications are undevelopable and/or the vast majority of targets are undruggable. We therefore asked what proportion of genetically supported G–I pairs had been developed to at least phase I, as a function of therapy area cross-tabulated against Open Targets predicted tractability status or membership in canonically 'druggable' protein families, using families from ref. 22 as well as UniProt pkinfam for kinases[36]. We also grouped at the level of gene, rather than G–I pair (Extended Data Fig. 8).

## Druggability and protein families

Antibody and small molecule druggability status was taken from Open Targets[37]. For antibody tractability, Clinical Precedence, Predicted Tractable–High Confidence and Predicted Tractable–Medium to Low Confidence were included. For small molecules, Clinical Precedence, Discovery Precedence and Predicted Tractable were included. Protein families were from sources described previously[22], plus the pkinfam kinase list from UniProt[36]. To make these lists non-overlapping, genes that were both kinases and also enzymes, ion channels or nuclear receptors were considered to be kinases only.

## Statistics

Analyses were conducted in R 4.2.0. For binomial proportions P(G) and P(S), error bars are Wilson 95% CIs, except for P(S) for phase I–launch for which the Wald method is used to compute the confidence intervals on the product of the individual probabilities of success at each phase. RS uses Katz 95% CIs, with the phase I launch RS based on the number of programs entering phase I and succeeding in phase III. Effects of continuous variables on probability of launch were assessed using logistic regression. Differences in RS between therapy areas were tested using the Cochran–Mantel–Haenszel chi-squared test (cmh.test from the R lawstat package, v.3.4). Pipeline progression of D–I pairs conditioned on the highest phase reached by a drug was modelled using an ordinal logit model (polr with Hess = TRUE from the R MASS package, v.7.3-56). Correlations across therapy areas were tested by weighted Pearson's correlation (wtd.cor from the R weights package, v.1.0.4); to control for the amount of data available in each therapy area, the number of genetically supported T–I pairs having reached at least phase I was used as the weight. Enrichments of T–I pairs in the utilization analysis were tested using Fisher's exact test. All statistical tests were two-sided.

## Reporting summary

Further information on research design is available in the Nature Portfolio Reporting Summary linked to this article.

## Data availability

An analytical dataset is provided at GitHub at https://github.com/ericminikel/genetic_support/ (ref. 38) and is sufficient to reproduce all figures and statistics herein. This repository is permanently archived at Zenodo at https://doi.org/10.5281/zenodo.10783210 (ref. 39). Source data are provided with this paper.

## Code availability

Source code is provided at GitHub at https://github.com/ericminikel/genetic_support/ (ref. 38) and is sufficient to reproduce all figures and statistics herein. This code is permanently archived at the Zenodo repository at https://doi.org/10.5281/zenodo.10783210 (ref. 39).

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

**Acknowledgements** This study was funded by Deerfield.

**Author contributions** M.R.N. and E.V.M. conceived and designed the study. E.V.M., J.L.P., C.C.D. and M.R.N. performed analyses. M.R.N. supervised the research. M.R.N. and E.V.M. drafted the manuscript. E.V.M., J.L.P., C.C.D. and M.R.N. reviewed and approved the final manuscript.

**Competing interests** M.R.N. is an employee of Deerfield and Genscience. C.C.D. is an employee of Deerfield. E.V.M. and J.L.P. are consultants to Deerfield. Unrelated to the current work, E.V.M. acknowledges speaking fees from Eli Lilly, consulting fees from Alnylam and research support from Ionis, Gate, Sangamo and Eli Lilly.

**Additional information**
**Correspondence and requests for materials** should be addressed to Matthew R. Nelson.

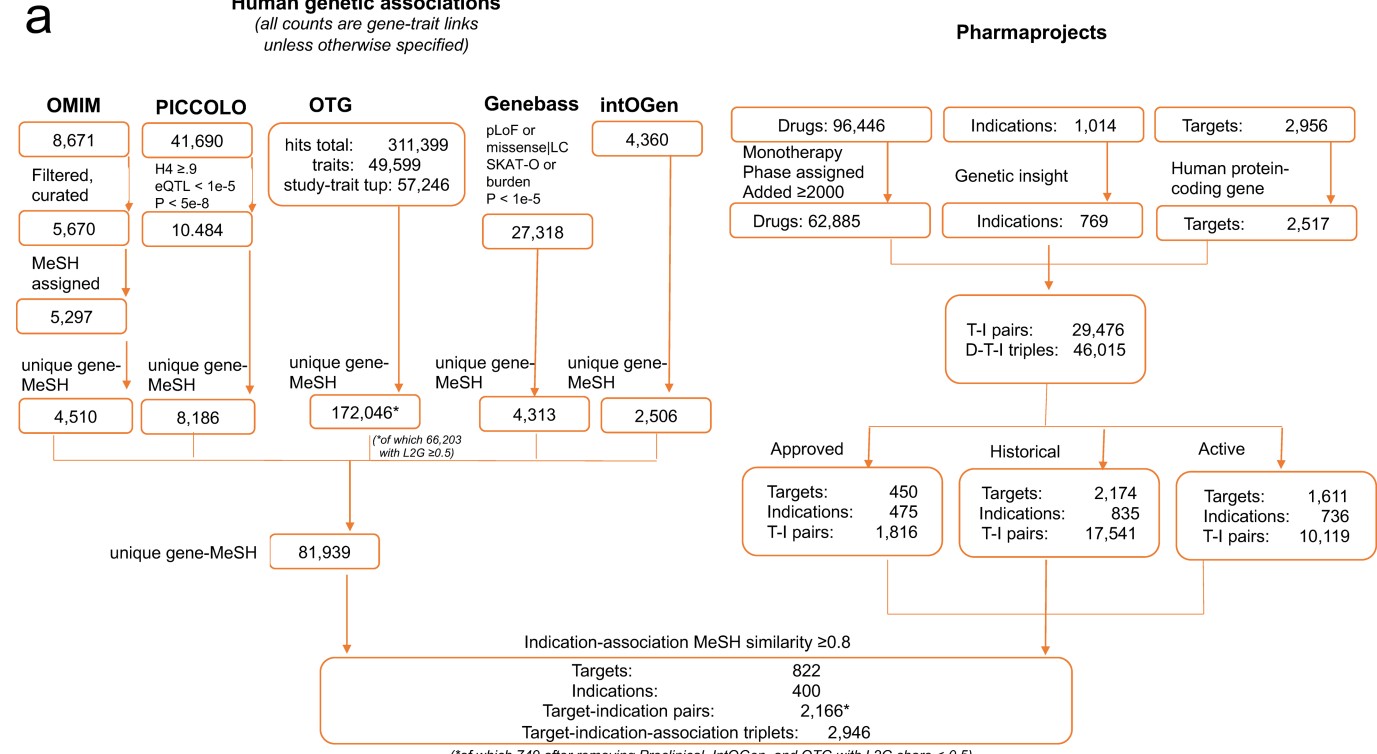

## a

**Human genetic associations**
*(all counts are gene-trait links unless otherwise specified)*

**Pharmaprojects**

| OMIM | PICCOLO | OTG | Genebass | intOGen |
|---|---|---|---|---|
| 8,671 | 41,690 | hits total: 311,399<br>traits: 49,599<br>study-trait tup: 57,246 | pLoF or missense\|LC SKAT-O or burden P < 1e-5<br>4,360 | |

OMIM: Filtered, curated → 5,670 → MeSH assigned → 5,297

PICCOLO: H4 ≥.9 eQTL < 1e-5 P < 5e-8 → 10,484

Genebass: 27,318

unique gene-MeSH:
- OMIM: 4,510
- PICCOLO: 8,186
- OTG: 172,046* (*of which 66,203 with L2G ≥0.5)
- Genebass: 4,313
- intOGen: 2,506

unique gene-MeSH: **81,939**

Pharmaprojects:
- Drugs: 96,446 → Monotherapy Phase assigned Added ≥2000 → Drugs: 62,885
- Indications: 1,014 → Genetic insight → Indications: 769
- Targets: 2,956 → Human protein-coding gene → Targets: 2,517

T-I pairs: 29,476
D-T-I triples: 46,015

| Approved | Historical | Active |
|---|---|---|
| Targets: 450<br>Indications: 475<br>T-I pairs: 1,816 | Targets: 2,174<br>Indications: 835<br>T-I pairs: 17,541 | Targets: 1,611<br>Indications: 736<br>T-I pairs: 10,119 |

**Indication-association MeSH similarity ≥0.8**
Targets: 822
Indications: 400
Target-indication pairs: 2,166*
Target-indication-association triplets: 2,946

*(*of which 749 after removing Preclinical, IntOGen, and OTG with L2G share < 0.5)*

## b

**drug programs**

| gene | indication MeSH ID | indication MeSH term | phase |
|---|---|---|---|
| *ABCC8* | D000070642 | Brain injury, traumatic | Phase II |
| *ABCC8* | D003924 | Diabetes Mellitus, Type 2 | Launched |
| *FFAR1* | D003924 | Diabetes Mellitus, Type 2 | Phase III |
| *IL1R1* | D003924 | Diabetes Mellitus, Type 2 | Phase II |

similarity = 1.0

**human genetic associations**

| gene | association MeSH ID | association MeSH term | source |
|---|---|---|---|
| *ABCC8* | D003924 | Diabetes Mellitus, Type 2 | OTG |
| *ABCC8* | D003924 | Diabetes Mellitus, Type 2 | OMIM |
| *ABCC8* | D007003 | Hypoglycemia | OMIM |
| *ABCC8* | D000428 | Alcohol Drinking | Genebass |
| *IL1R1* | D015212 | Inflammatory Bowel Diseases | OTG |

**Extended Data Fig. 1 | Data processing schematic. A)** Dataset size, filters, and join process for Pharmaprojects and human genetic evidence. Note that a drug can be assigned multiple targets, and can be approved for multiple indications. The entire analysis described herein has also been run restricted to only those drugs with exactly one target annotated (Figs. S1–S11). **B)** Illustration of the definition of genetic support. A table of drug development programs with one row per target-indication pair (left) is joined to a table of human genetic associations based on the identity of the gene encoding the drug target and the similarity between the drug indication MeSH term and the genetically associated trait MeSH term being ≥ 0.8. Drug program rows with a joined row in the genetic associations table are considered to have genetic support.

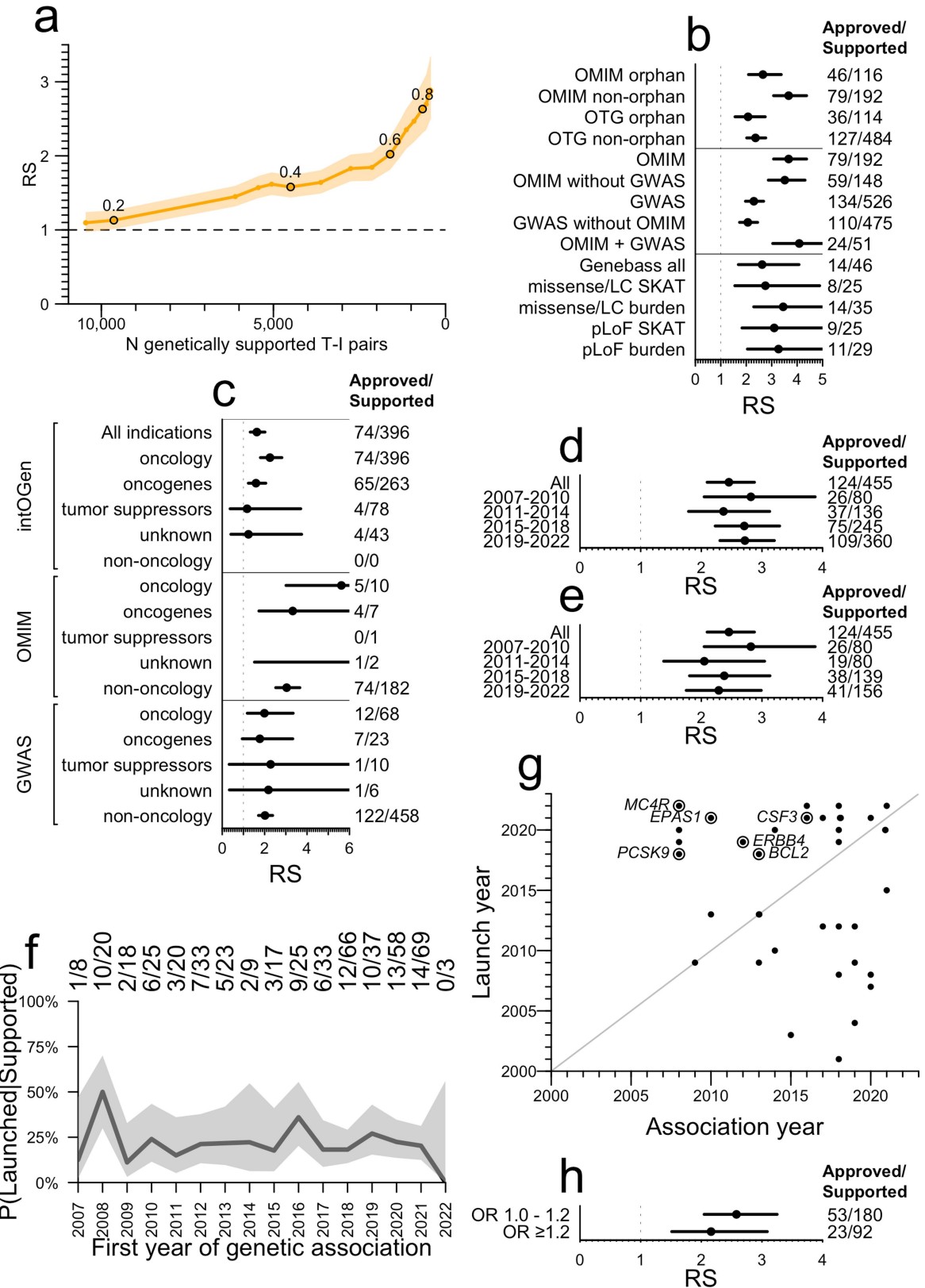

**Extended Data Fig. 2** | See next page for caption.

**Extended Data Fig. 2 | Further analysis of influence of characteristics of genetic associations on relative success. A**) Sensitivity of RS to the similarity threshold between the MeSH ID for the genetically associated trait and the MeSH ID for the clinically developed indication. The threshold is varied by units of 0.05 (labels) and the results are plotted as RS (y axis) versus number of genetically supported T-I pairs (x axis). **B**) Breakdown of OTG and OMIM RS values by whether any drug for each T-I pair has had orphan status assigned. The N of genetically supported T-I pairs (denominator) and, of those, launched T-I pairs (numerator) is shown at right. Values for the full 2×2 contingency table including the non-supported pairs, used to calculate RS, are provided in Table S12. Total N = 13,022 T-I pairs, of which 3,149 are orphan. The center is the RS point estimate and error bars are Katz 95% confidence intervals. **C**) RS for somatic genetic evidence from IntOGen versus germline genetic evidence, for oncology and non-oncology indications. Note that the approved/supported proportions displayed for the top two rows are identical because all IntOGen genetic support is for oncology indications, yet the RS is different because the number of non-supported approved and non-supported clinical stage programs is different. In other words, in the "All indications" row, there is a Simpson's paradox that diminishes the apparent RS of IntOGen – IntOGen support improves success rate (see 2nd row) but also selects for oncology, an area with low baseline success rate (as shown in Extended Data Fig. 6a). N is displayed at right as in (B), with full contingency tables in Table S13. Total N = 13,022 T-I pairs, of which 6,842 non-oncology, 6,180 oncology, 1,287 targeting IntOGen oncogenes, 284 targeting tumor suppressors, and 176 targeting IntOGen genes of unknown mechanism. The center is the RS point estimate and error bars are Katz 95% confidence intervals. **D**) As for top panel of Fig. 1d, but without removing replications or OMIM-supported T-I pairs. N is displayed as in (B), with full contingency tables in Table S14. Total N = 13,022 T-I pairs. The center is the RS point estimate and error bars are Katz 95% confidence intervals. **E**) As for top panel of Fig. 1d, removing replications but not removing OMIM-supported T-I pairs. N is displayed as in (B), with full contingency tables in Table S15. Total N = 13,022 T-I pairs. The center is the RS point estimate and error bars are Katz 95% confidence intervals. **F**) Proportion of T-I pairs supported by a GWAS Catalog association that are launched (versus phase I-III) as a function of the year of first genetic association. **G**) Launched T-I pairs genetically supported by OTG GWAS, shown by year of launch (y axis) and year of first genetic association (x axis). Gene symbols are labeled for first approvals of targets with at least 5 years between association and launch. Of 104 OTG-supported launched T-I pairs (Fig. 1d), year of drug launch was available for N = 38 shown here, of which 18 (47%) acquired genetic support only in or after the year of launch. The true proportion of launched T-I whose GWAS support is retrospective may be larger if the T-I with a missing launch year are more often older drug approvals less well annotated in Pharmaprojects. **H**) Lack of impact of GWAS Catalog lead SNP odds ratio (OR) on RS when using the same OR breaks as used by King et al.[15]. N is displayed as in (B), with full contingency tables in Table S18. Total N = 13,022 T-I pairs. The center is the RS point estimate and error bars are Katz 95% confidence intervals. See Fig. S4 for the same analyses restricted to drugs with a single known target.

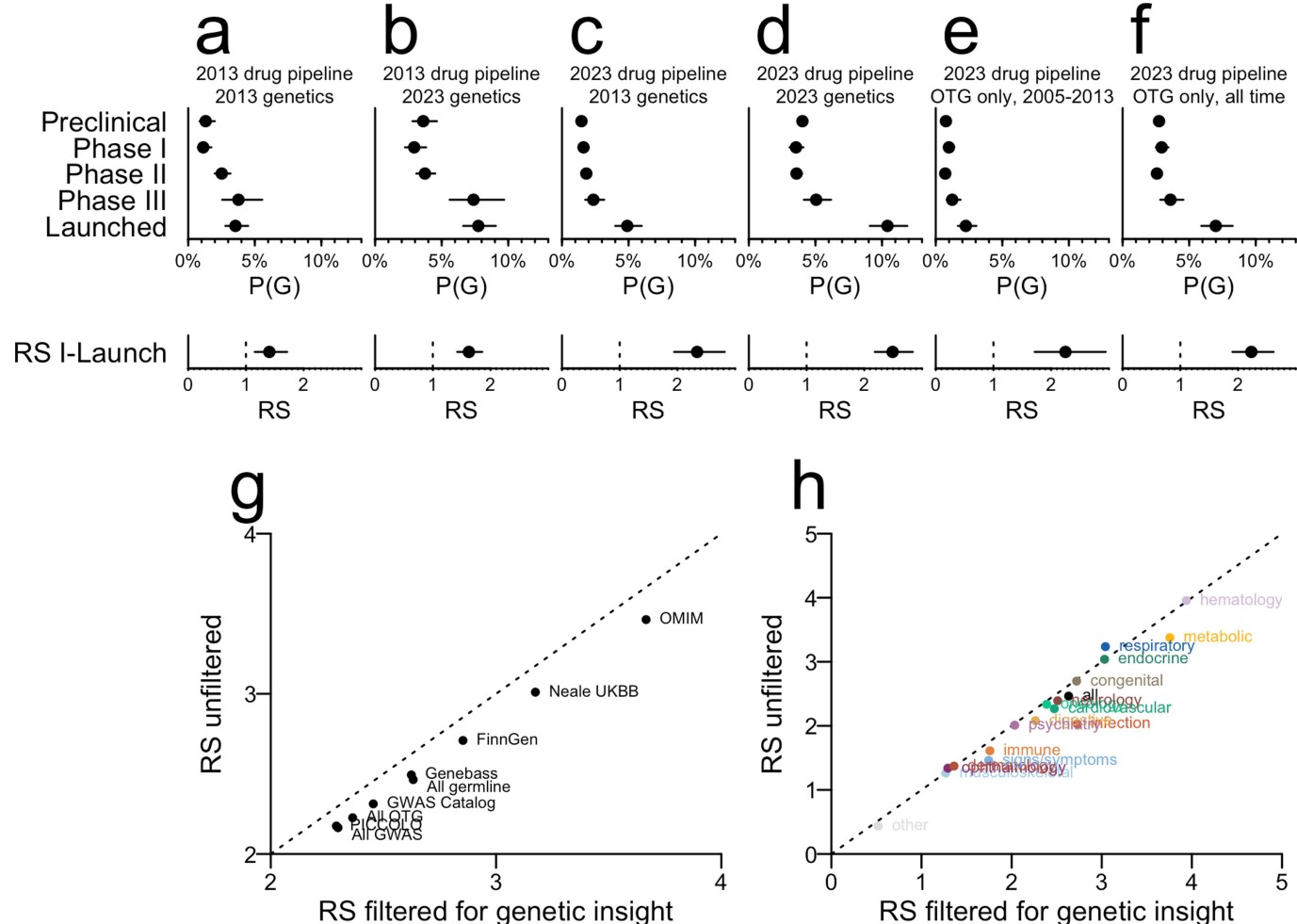

**Extended Data Fig. 3 | Sensitivity to changes in genetic data and drug pipeline over the past decade and to the 'genetic insight' filter.** "2013" here indicates the data freezes from Nelson et al.[5] (that study's supplementary dataset 2 for genetics and supplementary dataset 3 for drug pipeline); "2023" indicates the data freezes in the present study. All datasets were processed using the current MeSH similarity matrix, and because "genetic insight" changes over time (more traits have been studied genetically now than in 2013), all panels are unfiltered for genetic insight (hence numbers in panel D differ from those in Fig. 1a). Every panel shows the proportion of combined (both historical and active) target-indication pairs with genetic support, or P(G), by development phase. **A)** 2013 drug pipeline and 2013 genetics. **B)** 2013 drug pipeline and 2023 genetics. **C)** 2023 drug pipeline and 2013 genetics. **D)** 2023 drug pipeline and 2023 genetics. **E)** 2023 drug pipeline with only OTG GWAS hits through 2013 and no other sources of genetic evidence. **F)** 2023 drug pipeline with only OTG GWAS hits for all years, no other sources of genetic evidence. We note that the increase in P(G) over the past decade[5] is almost entirely attributable to new genetic evidence (e.g. contrast B vs. A, D vs. C, F vs. E) rather than changes in the drug pipeline (e.g. compare A vs. C, B vs. D). In contrast, the increase in RS is due mostly to changes in the drug pipeline (compare C, D, E, F vs. A, B), in line with theoretical expectations outlined by

Hingorani et al.[16] and consistent with the findings of King et al.[15] We note that both the contrasts in this figure, and the fact that genetic support is so often retrospective (Extended Data Fig. 2g) suggest that P(G) will continue to rise in coming years. For 2013 drug pipeline, N = 8,624 T-I pairs (1,605 preclinical, 1,772 phase I, 2,779 phase II, 636 phase III, and 1,832 launched); for 2023 drug pipeline, N = 29,464 T-I pairs (N = 12,653 preclinical, 4,946 phase I, 8,268 phase II, 1,781 phase III, and 1,816 launched). Details including numerator and denominator for P(G) and full contingency tables for RS are provided in Tables S19 - S20. In A-F, the center is exact proportion and error bars are Wilson binomial 95% confidence intervals. Because all panels here are unfiltered for genetic insight, we also show the difference in RS across **G)** sources of genetic evidence and **H)** therapy areas when this filter is removed. In general, removing this filter decreases RS by 0.17; this varies only slightly between sources and areas. The largest impact is seen in Infection, where removing the filter drops the RS from 2.73 to 2.03. The relatively minor impact of removing the genetic insight filter is consistent with the findings of King et al.[15], who varied the minimum number of genetic associations required for an indication to be included, and found that risk ratio for progression (i.e. RS) was slightly diminished when the threshold was reduced. See Fig. S5 for the same analyses restricted to drugs with a single known target.

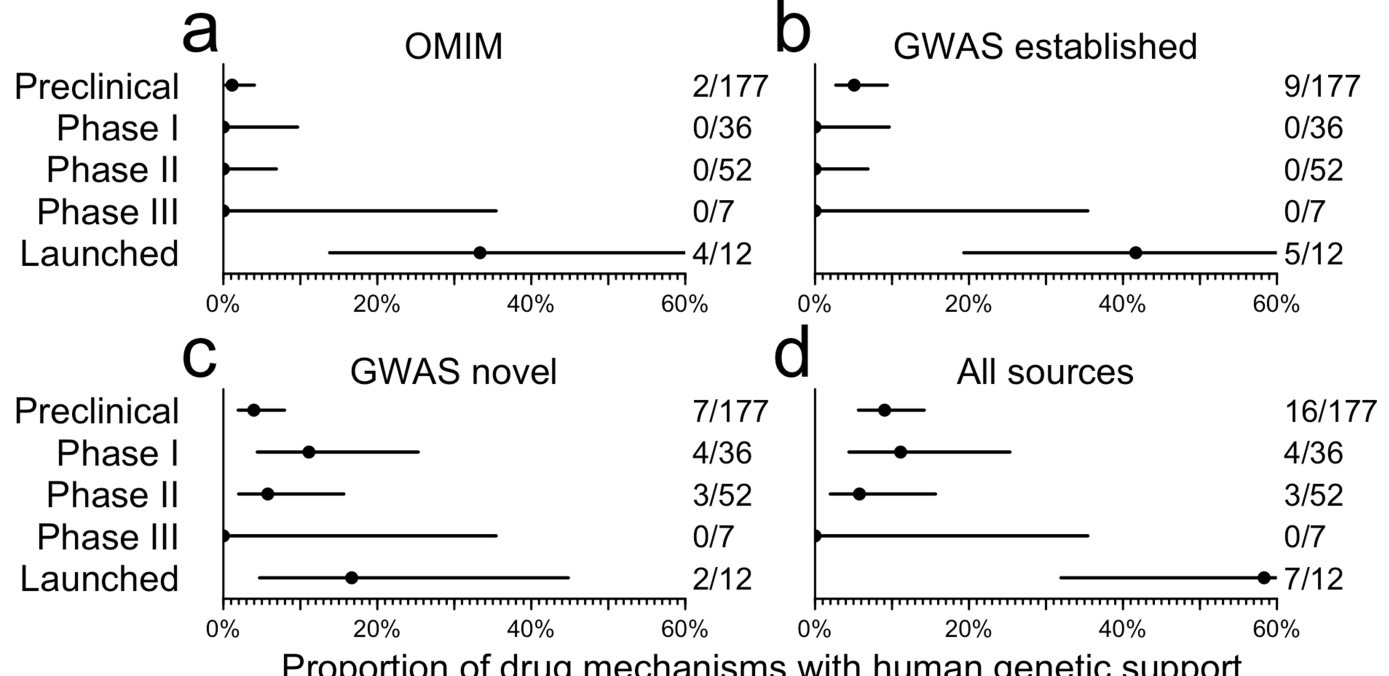

**Extended Data Fig. 4 | Proportion of type 2 diabetes drug targets with human genetic support by highest phase reached.** A) OMIM, B) established (2019 and earlier) GWAS genes, C) novel (new in Vujkovic 2020 or Suzuki 2023) GWAS genes, or D) any of the above. See Methods for details on GWAS dataset processing. N is indicated at right of each panel, with denominator being the number of T2D targets at each stage and the numerator being the number of those that are genetically supported. Total N = 284 targets. The center is the exact proportion and error bars are Wilson binomial 95% confidence intervals.

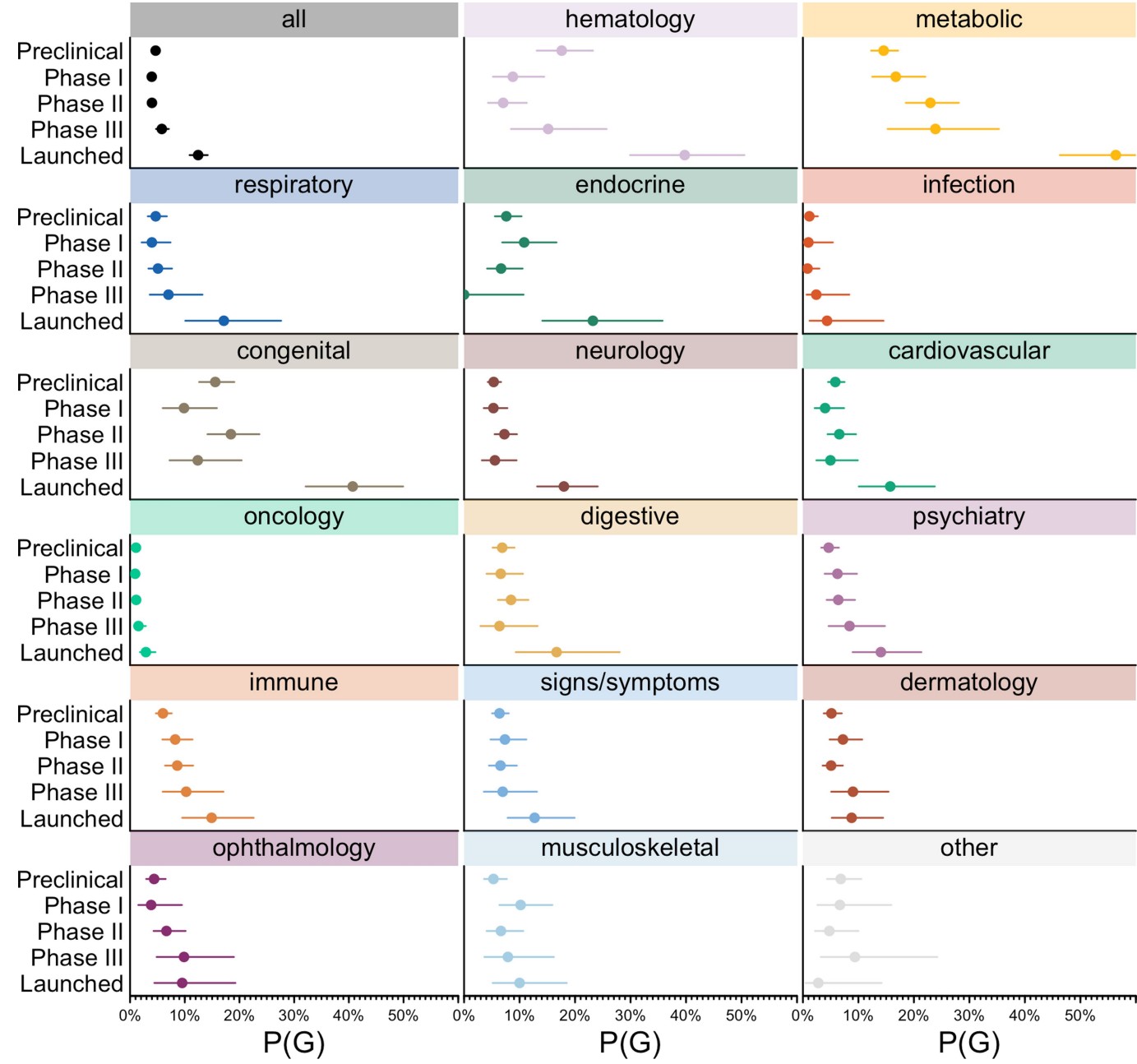

**Extended Data Fig. 5 | P(G) by phase versus therapy area.** Each panel represents one therapy area, and shows the proportion of target-indication pairs in that area with genetic support, or P(G), by development phase. The genetically supported and total number of T-I pairs at each phase in each therapy area are provided in Table S33. Total number of T-I pairs in any area: N = 10,839 preclinical, N = 4,421 phase I, N = 7,383 phase II, N = 1,551 phase III, N = 1,519 launched. The center is the exact proportion and error bars are Wilson binomial 95% confidence intervals. See Fig. S6 for the same analyses restricted to drugs with a single known target.

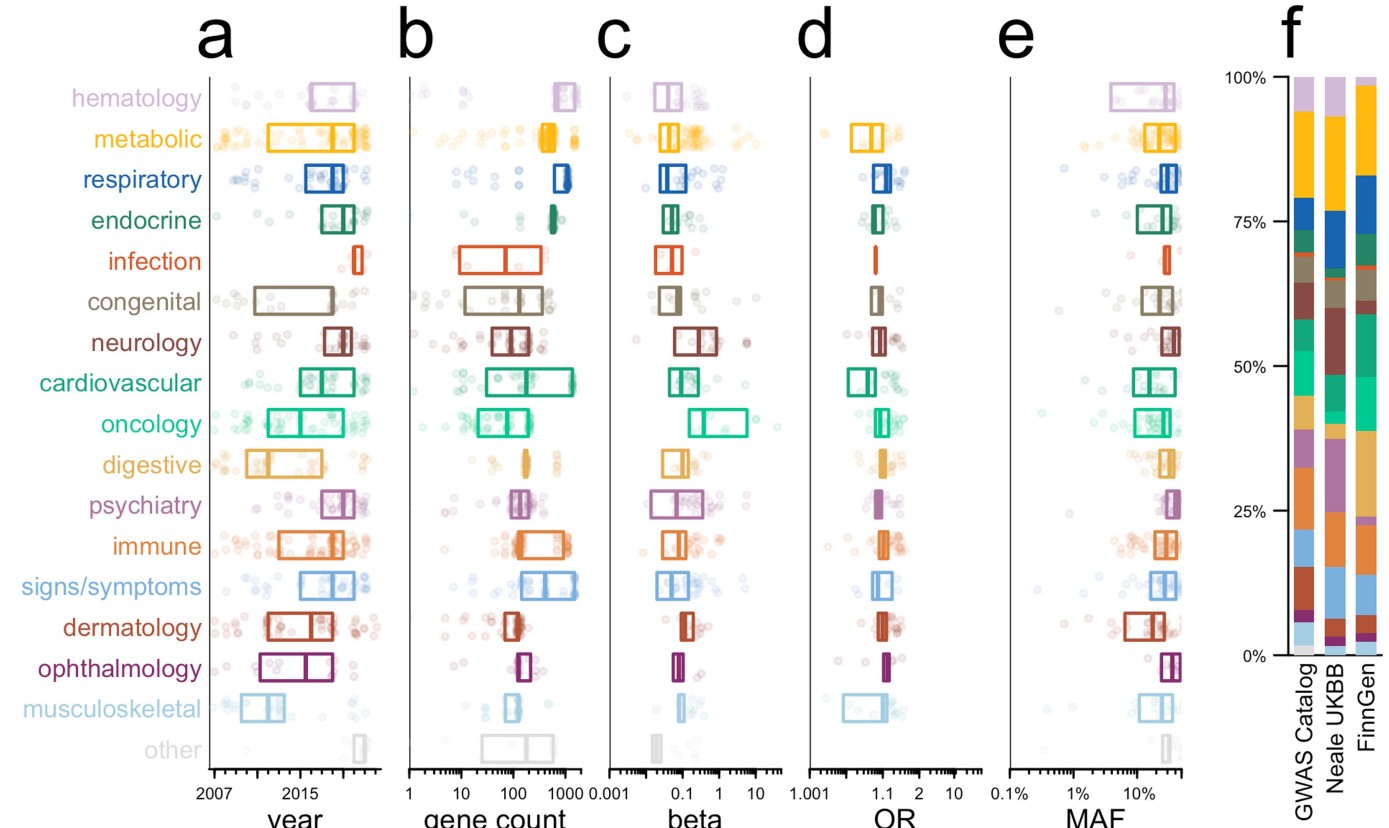

**Extended Data Fig. 6 | Confounding between therapy areas and properties of supporting genetic evidence.** In panels A-E, each point represents one GWAS Catalog-supported T-I pair in phase I through launched, and boxes represent medians and interquartile ranges (25th, 50th, and 75th percentile). Each panel A-E represents the cross-tabulation of therapy areas versus the properties examined in Fig. 1d. Kruskal-Wallis tests treat each variable as continuous, while chi-squared tests are applied to the discrete bins used in Fig. 1d. **A**) Year of discovery, Kruskal-Wallis P = 1.1e-11, chi-squared P = 2.9e-16, N = 686 target-indication-area (T-I-A) triplets; **B**) gene count, Kruskal-Wallis P = 6.2e-35, chi-squared P = 7.1e-47, N = 770 T-I-A triplets; **C**) absolute beta, Kruskal-Wallis P = 1.2e-5, chi-squared P = 1.7e-7, N = 461 T-I-A triplets; **D**) absolute odds ratio, Kruskal-Wallis P = 2.5e-5, chi-squared P = 4.3e-6, N = 305 T-I-A triplets; **E**) minor allele frequency, Kruskal-Wallis P = 5.7e-4, chi-squared P = 4.3e-3, N = 584 T-I-A triplets; **F**) Barplot of therapy areas of genetically supported T-I by source of GWAS data within OTG, chi-squared P = 2.4e-7. See Fig. S7 for the same analyses restricted to drugs with a single known target.

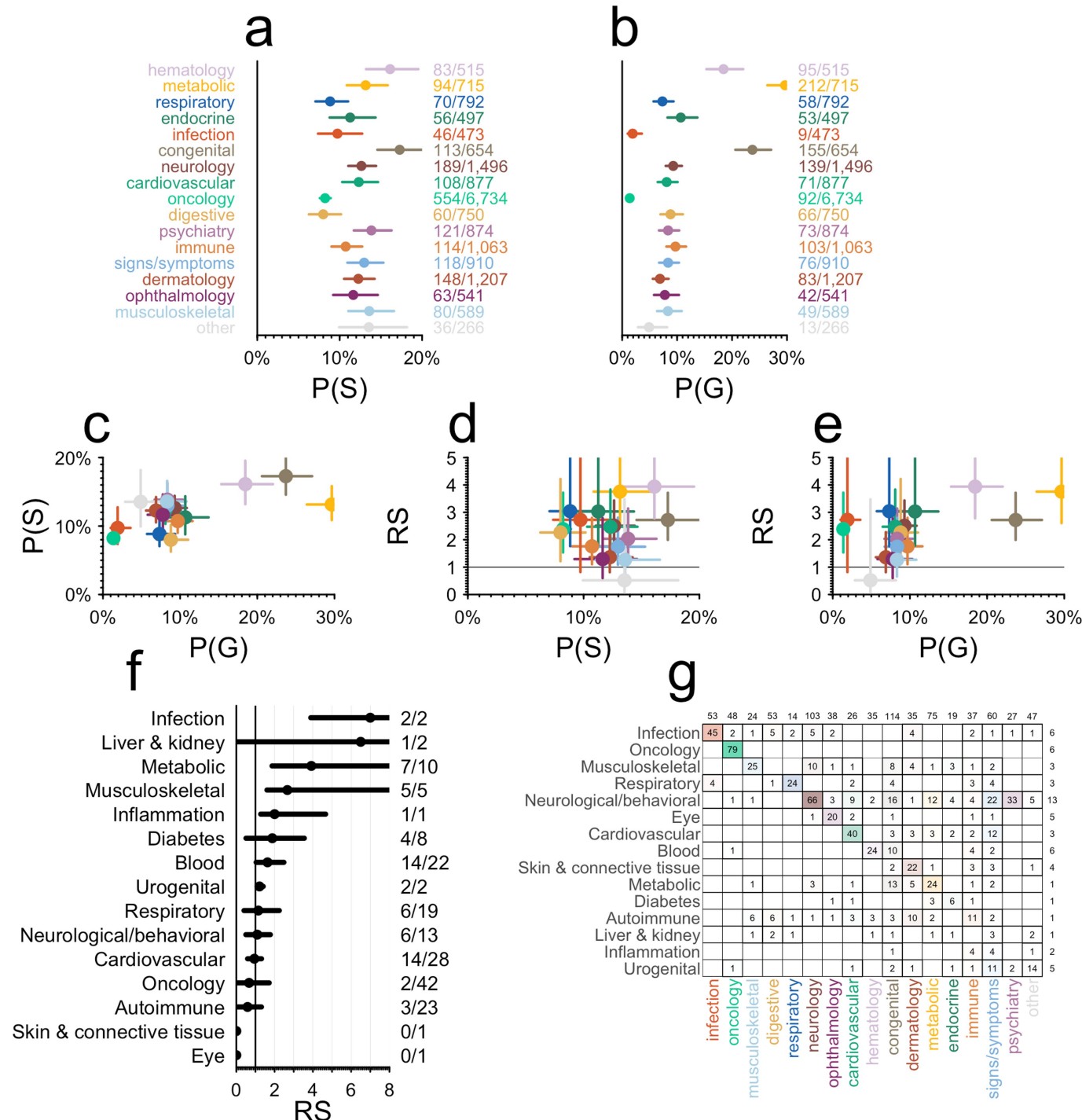

**Extended Data Fig. 7 | Further analyses of differences in relative success among therapy areas. A)** Probability of success, P(S), by therapy area, with Wilson 95% confidence intervals. The N shown at right indicates the number of launched T-I pairs (numerator) and number of T-I pairs reaching at least phase I (denominator). The center is the exact proportion and error bars are Wilson binomial 95% confidence intervals. **B)** Probability of genetic support, P(G), by therapy area, with Wilson 95% confidence intervals. The N shown at right indicates the number of genetically supported T-I pairs reaching at least phase I (numerator) and total number of T-I pairs reaching at least phase I (denominator). The center is the exact proportion and error bars are Wilson binomial 95% confidence intervals. **C)** P(S) vs. P(G), **D)** RS s. P(S), and **E)** RS vs. P(G) across therapy areas, with centers indicating point estimates and crosshairs representing 95% confidence intervals on both dimensions – Katz for RS and

Wilson for P(G) and P(S). For A-E, total N = 13,022 unique T-I pairs, but because some indications belong to > 1 therapy area, N = 16,900 target-indication-area (T-I-A) triples. For exact N and full contingency tables, see Table S28. **F)** Re-analysis of RS (x axis) broken down by therapy area using data from supplementary table 6 of Nelson et al.[5]. **G)** Confusion matrix showing the categorization of unique drug indications into therapy areas in Nelson et al.[5] versus current. Note that the current categorization is based on each indication's position in the MeSH ontological tree and one indication can appear in > 1 area, see Methods for details. Marginals along the top edge are the number of drug indications in each current therapy area that were absent from the 2015 dataset. Marginals along the right edge are the number of drug indications in each 2015 therapy area that are absent from the current dataset. See Fig. S8 for the same analyses restricted to drugs with a single known target.

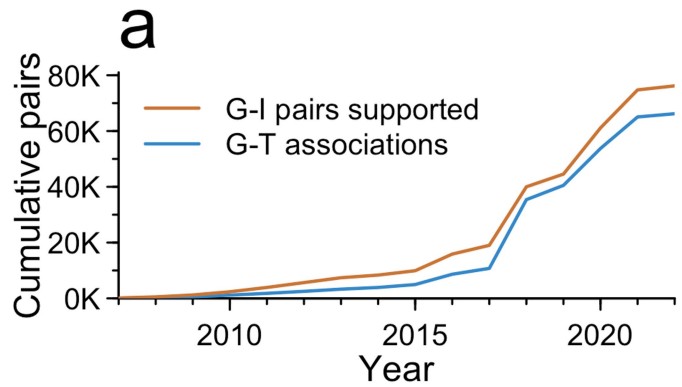

**a**

Cumulative pairs (0K–80K) vs Year (2010, 2015, 2020)
- G-I pairs supported
- G-T associations

**b**

| | all genes | predicted Ab tractable | predicted SM tractable | rhodopsin-like GPCRs | nuclear receptors | enzymes | ion channels | kinases |
|---|---|---|---|---|---|---|---|---|
| **all areas** | 373/10424 **3.6%** | 298/5292 **5.6%** | 302/3213 **9.4%** | 25/175 **14%** | 10/34 **29%** | 67/597 **11%** | 24/210 **11%** | 35/239 **15%** |
| **hematology** | 37/3072 **1.2%** | 30/1514 **2%** | 32/1046 **3.1%** | 2/51 **3.9%** | 0/13 **0%** | 8/186 **4.3%** | 0/35 **0%** | 3/87 **3.4%** |
| **metabolic** | 89/3914 **2.3%** | 72/2052 **3.5%** | 69/1318 **5.2%** | 5/52 **9.6%** | 6/20 **30%** | 11/249 **4.4%** | 1/53 **1.9%** | 2/104 **1.9%** |
| **respiratory** | 17/691 **2.5%** | 16/368 **4.3%** | 11/238 **4.6%** | 2/6 **33%** | 0/4 **0%** | 2/42 **4.8%** | 1/18 **5.6%** | 0/10 **0%** |
| **endocrine** | 28/2033 **1.4%** | 25/1016 **2.5%** | 24/667 **3.6%** | 5/36 **14%** | 2/22 **9.1%** | 1/138 **0.72%** | 1/40 **2.5%** | 3/55 **5.5%** |
| **infection** | 3/222 **1.4%** | 3/129 **2.3%** | 1/79 **1.3%** | 0/4 **0%** | 0/2 **0%** | 0/13 **0%** | 0/6 **0%** | 0/2 **0%** |
| **congenital** | 84/3555 **2.4%** | 62/1912 **3.2%** | 61/1311 **4.7%** | 2/38 **5.3%** | 2/19 **11%** | 15/236 **6.4%** | 1/80 **1.2%** | 2/87 **2.3%** |
| **neurology** | 72/2810 **2.6%** | 61/1531 **4%** | 56/957 **5.9%** | 6/44 **14%** | 0/8 **0%** | 10/164 **6.1%** | 10/96 **10%** | 3/67 **4.5%** |
| **cardiovascular** | 33/1706 **1.9%** | 31/899 **3.4%** | 30/578 **5.2%** | 2/22 **9.1%** | 1/10 **10%** | 9/122 **7.4%** | 8/48 **17%** | 1/62 **1.6%** |
| **oncology** | 113/2045 **5.5%** | 83/949 **8.7%** | 99/716 **14%** | 5/26 **19%** | 2/13 **15%** | 24/130 **18%** | 0/33 **0%** | 28/79 **35%** |
| **digestive** | 31/1610 **1.9%** | 25/871 **2.9%** | 21/552 **3.8%** | 3/33 **9.1%** | 2/12 **17%** | 6/95 **6.3%** | 1/21 **4.8%** | 2/51 **3.9%** |
| **psychiatry** | 30/1663 **1.8%** | 28/891 **3.1%** | 29/524 **5.5%** | 8/34 **24%** | 0/9 **0%** | 4/101 **4%** | 6/55 **11%** | 0/49 **0%** |
| **immune** | 44/1566 **2.8%** | 41/852 **4.8%** | 29/545 **5.3%** | 4/30 **13%** | 0/4 **0%** | 9/95 **9.5%** | 0/19 **0%** | 2/49 **4.1%** |
| **signs/symptoms** | 45/2659 **1.7%** | 39/1408 **2.8%** | 39/842 **4.6%** | 7/55 **13%** | 4/19 **21%** | 4/155 **2.6%** | 10/73 **14%** | 3/71 **4.2%** |
| **dermatology** | 37/1590 **2.3%** | 33/869 **3.8%** | 23/522 **4.4%** | 1/18 **5.6%** | 0/6 **0%** | 6/106 **5.7%** | 0/24 **0%** | 3/45 **6.7%** |
| **ophthalmology** | 21/1573 **1.3%** | 16/888 **1.8%** | 14/478 **2.9%** | 1/16 **6.2%** | 1/12 **8.3%** | 3/87 **3.4%** | 2/44 **4.5%** | 0/43 **0%** |
| **musculoskeletal** | 27/1359 **2%** | 24/735 **3.3%** | 19/449 **4.2%** | 1/15 **6.7%** | 2/5 **40%** | 2/82 **2.4%** | 1/24 **4.2%** | 4/42 **9.5%** |
| **other** | 6/1186 **0.51%** | 5/586 **0.85%** | 5/363 **1.4%** | 0/19 **0%** | 1/7 **14%** | 0/67 **0%** | 0/28 **0%** | 1/30 **3.3%** |

**Extended Data Fig. 8 | Level of utilization of genetic support among targets.** As for Fig. 3, but grouped by target instead of T-I pair. Thus, the denominator for each cell is the number of targets with at least one genetically supported indication, and each target counts towards the numerator if at least one genetically supported indication has reached phase I. See Fig. S9 for the same analyses restricted to drugs with a single known target.

# Reporting Summary

## Statistics

For all statistical analyses, confirm that the following items are present in the figure legend, table legend, main text, or Methods section.

| n/a | Confirmed | |
|---|---|---|
| ☐ | ☒ | The exact sample size (*n*) for each experimental group/condition, given as a discrete number and unit of measurement |
| ☒ | ☐ | A statement on whether measurements were taken from distinct samples or whether the same sample was measured repeatedly |
| ☐ | ☒ | The statistical test(s) used AND whether they are one- or two-sided *Only common tests should be described solely by name; describe more complex techniques in the Methods section.* |
| ☐ | ☒ | A description of all covariates tested |
| ☐ | ☒ | A description of any assumptions or corrections, such as tests of normality and adjustment for multiple comparisons |
| ☐ | ☒ | A full description of the statistical parameters including central tendency (e.g. means) or other basic estimates (e.g. regression coefficient) AND variation (e.g. standard deviation) or associated estimates of uncertainty (e.g. confidence intervals) |
| ☐ | ☒ | For null hypothesis testing, the test statistic (e.g. *F*, *t*, *r*) with confidence intervals, effect sizes, degrees of freedom and *P* value noted *Give P values as exact values whenever suitable.* |
| ☒ | ☐ | For Bayesian analysis, information on the choice of priors and Markov chain Monte Carlo settings |
| ☒ | ☐ | For hierarchical and complex designs, identification of the appropriate level for tests and full reporting of outcomes |
| ☐ | ☒ | Estimates of effect sizes (e.g. Cohen's *d*, Pearson's *r*), indicating how they were calculated |

*Our web collection on statistics for biologists contains articles on many of the points above.*

## Software and code

Policy information about availability of computer code

| Data collection | No software was used to collect data. |
|---|---|
| Data analysis | R 4.2.0 scripts used for data analysis have been made publicly available, see code availability statement. R package dependencies and the versions used are as follows: tidyverse_1.3.1, janitor_2.1.0, binom_1.1-1.1, glue_1.6.2, lawstat_3.4, weights_1.0.4, epitools_0.5-10.1, DescTools_0.99.45, openxlsx_4.2.5, optparse_1.7.1, MASS_7.3-56. |

For manuscripts utilizing custom algorithms or software that are central to the research but not yet described in published literature, software must be made available to editors and reviewers. We strongly encourage code deposition in a community repository (e.g. GitHub). See the Nature Portfolio guidelines for submitting code & software for further information.

## Data

Policy information about availability of data

All manuscripts must include a data availability statement. This statement should provide the following information, where applicable:

- Accession codes, unique identifiers, or web links for publicly available datasets
- A description of any restrictions on data availability
- For clinical datasets or third party data, please ensure that the statement adheres to our policy

Data availability. An analytical dataset is provided at https://github.com/ericminikel/genetic_support/ and is sufficient to reproduce all figures and statistics herein.

## Human research participants

Policy information about studies involving human research participants and Sex and Gender in Research.

| Reporting on sex and gender | N/A |
|---|---|
| Population characteristics | N/A |
| Recruitment | N/A |
| Ethics oversight | N/A |

Note that full information on the approval of the study protocol must also be provided in the manuscript.

# Field-specific reporting

Please select the one below that is the best fit for your research. If you are not sure, read the appropriate sections before making your selection.

☒ Life sciences ☐ Behavioural & social sciences ☐ Ecological, evolutionary & environmental sciences

For a reference copy of the document with all sections, see nature.com/documents/nr-reporting-summary-flat.pdf

# Life sciences study design

All studies must disclose on these points even when the disclosure is negative.

| Sample size | No pre-determined sample size was chosen; we used all available drug development data since 2000. |
|---|---|
| Data exclusions | We focused on the time period since 2000 because data are relatively complete for this time period. |
| Replication | Our study relies on retrospective analysis of drug development successes and failures worldwide since 2000, worldwide. Because human history has only occurred once, there is no opportunity for replication per se. Instead, to provide robustness, we performed each analysis under multiple different filters and conditions, for example, we re-ran the entire analysis restricting to only drugs with exactly one target, we examined the contribution of different genetic data sources, we varied our quantitative filtering thresholds, and so on. Core findings of our study were robust to all different conditions examined. |
| Randomization | Our study did not involve individual human or animal participants. Historical data on drug development successes and failures are inherently retrospective and cannot be randomized. |
| Blinding | Our study was retrospective in nature and there was no opportunity for prospective blinding. We note that many of the drug development programs analyzed here were undertaken before the discovery of relevant human genetic associations, and thus the pharmaceutical companies involved in these programs were in a sense "blinded", however, no formal blinding was undertaken. |

# Reporting for specific materials, systems and methods

We require information from authors about some types of materials, experimental systems and methods used in many studies. Here, indicate whether each material, system or method listed is relevant to your study. If you are not sure if a list item applies to your research, read the appropriate section before selecting a response.

### Materials & experimental systems

| n/a | Involved in the study |
|---|---|
| ☒ ☐ | Antibodies |
| ☒ ☐ | Eukaryotic cell lines |
| ☒ ☐ | Palaeontology and archaeology |
| ☒ ☐ | Animals and other organisms |
| ☒ ☐ | Clinical data |
| ☒ ☐ | Dual use research of concern |

### Methods

| n/a | Involved in the study |
|---|---|
| ☒ ☐ | ChIP-seq |
| ☒ ☐ | Flow cytometry |
| ☒ ☐ | MRI-based neuroimaging |

