## [Peer Review File · Nature]

Manuscript Title: Refining the impact of genetic evidence on clinical success

Reviewer Comments & Author Rebuttals

Reviewer Reports on the Initial Version:

Referees' comments:

Referee #1 (Remarks to the Author):

Minikel et al are following up to Nelson et al's original seminal work from 2015 that clearly demonstrated the value of human genetics for target discovery. This work and the studies that followed it, certainly under-pinned further investments from pharma companies such as GSK, AZ, Pfizer, Lilly, Regeneron etc in human genetics.

Here Nelson follows up with a more systematic approach to evaluating human genetic support and compare across different therapeutic areas, addressing genetic effect size relationship to therapeutic impact and the importance of genetics by development phase. It's interesting work that I believe will be helpful for the field. I have a few suggestions.

1. The title is very broad but I am not convinced that the paper, is as broad as the title suggests. Fundamentally, most pharma now buy into the relevance of human genetics for research purposes, particularly for target ID. But two important areas that pharma are using human genetics for already that are not well addressed in the current manuscript are safety and Biomarker ID.. Amgen bought DECODE in about 2012 with the primary motive of using genetics to identify potential adverse effects. With the emergence of high throughput molecular data and longitudinal health data, married together with genetics, pharma are also using genetics for Biomarker discovery. While I appreciate that this is a lot of extra effort to fold into the manuscript, and is likely not possible, perhaps the authors could at least comment to these subjects also?

2. The paper would benefit from much more clarity of description. Currently, probably in a bid to reduce word count, it takes some effort to understand. E.g. RS is mentioned lines 17&18, described in lines 44/45 and defined in lines 344, where the elements in the definition are not given e.g. what is X?

2b. The T2D MVP paper is described but what is meant by novel is not clearly defined is it novel in that a gene has not been characterised before. Is it novel as genetics has not shown an association with T2D before. Is it novel because genetics has not shown an association with T2D nor glycemic traits?. Is it novel because it's not been associated with T2D, glycemic traits, and is not known biology/pathway?

2c. Human gene target is used a lot, but most therapies still target proteins that are encoded by genes. Are the authors restricting to gene targets? 2d. Indication-trait similarity threshold is used line 48 but what is it? 2e. Could the authors review for clarity the whole MS.

3. Stats description should be improved for clarity and justification of significance thresholds

provided. Including for PICCOLO - Why so lenient a threshold for eQTL vs GWAS?

4. Lack of description of the data sources e.g. the link to the website (ref 18) re-directs to citeline.com and not the website with any browsable data. What is in citeline? Is it all submissions to FDA, or are other medicines authorities also included?

5. I would like the authors to be really crisp on how they define human genetics support. It's not entirely clear to me how this is done. Perhaps a schema could be included as a main figure as this is key to the conclusions drawn? Are all evidences treated equally?

Minor comments

Line 30/31 - that GWAS are less predictive of clinical success than Mendelian disease is not surprising given that the causal gene is known for Mendelian disease, while GWAS only points to a region of the genome. But we are not comparing the same thing and there is more to it than this simple comparison. There is considerable investment in human genetics by pharma and decisions are now being made based on human genetics in the pharma setting eg targets being stopped due to lack of human evidence or potential safety suggested by genetics.

Line 342 what are G and S in P(G) P(S)?

Some of the databases have been accessed 2-3 years ago eg intogen and OMIM. Why not more current? OMIM has been updated. The stats on numbers of gene-phenotype associations look quite different as -provided by OMIM in current release vs quoted in the paper. For others eg open targets, when the database was accessed is completely missing.

The omni-genie model discussion is a bit of a distraction and doesn't add to the story. I think all the authors are trying to say is that it's difficult to go from list of 100s of causal genes to a short prioritised list of potential targets, which is not controversial.

For locus to gene share - is this defined per disease or is it across diseases? Can this be clarified line 268.

The authors might like to comment on the lag time between genetic discovery and clinical translation. This is an important feature as the genetic targets that are identified today won't reach first human dose for likely 5-10 years. Therefore we will see a continued growth in targets that have genetic support given the large continuing investment in genetics cohorts by pharma eg ukb, finngen

Referee #2 (Remarks to the Author):

In this manuscript, the authors convincingly demonstrate that drugs directed against targets with human genetic support have a higher probability of regulatory approval. Using the latest genetic datasets they refine earlier probability of success estimates for a range of therapeutic areas, explore which genetic mechanisms are more versus less likely to differentiate successful from unsuccessful drugs under development and conclude that thus far only a minority of genetically supported target-indication pairs have been leveraged for drug discovery.

While the conclusions from this study are not fundamentally novel, this is an important and timely piece of work that builds upon and expands earlier studies exploring the potential of human genetics to reduce the horrendous failure rates in drug development. The study is elegantly designed, technically straightforward and the statistics and conclusions seem appropriate.

1. In their seminal 2015 paper the authors provided evidence that the fraction of genetic targets increases with pipeline stage (i.e., approved > Ph III > Ph II > Ph I). Based on their new analysis it now does no longer seem to hold up that genetic support in general reduces the risk of failure to progress to the next development phase, with modest separation starting only between Ph II and Ph III and a substantial separation not before Ph III and Launched (Fig1A). The earlier observation though still seems to hold up for drugs with one target only (FigS6) as well as for distinct therapeutic areas (Fig2A-F). If this interpretation is correct, this refinement should be stated and discussed in the manuscript.

2. Notably, $P(G)$ appears higher for all categories and phases in the current paper than in the authors' previous paper. For instance, in 2015 $P(G)$ (if equivalent) for Ph I was ~2% for OMIM+GWASdb while now it is ~4%, even for the historical category. Is this primarily a reflection that thanks to growing genetic datasets we can now just find more genetic support for existing Development targets? Or does it also come from an increased number of new genetic entries into the Development space? (Or could it simply be because the definition of genetic support is slightly different between the two papers?) More granularity on which factors may have doubled $P(G)$ since 2015, and in which phases this effect is most prominent, would be appreciated.

3. In the authors' 2015 analysis, "musculoskeletal" had been the therapeutic area with the highest enrichment for genetic targets among approvals. In the current paper this therapeutic area shows the lowest RS across all phases. Conversely, "metabolic" stays near the top in both analyses. This still seems to be the case after re-analysis of the 2015 data for RS (FigS4F). What explains such disparities? Is it only because the "search space" (i.e. available genetic data) has been considerably growing in some indications (e.g. metabolic; Fig2F) while for others (e.g. musculoskeletal) not so much. In other words: Is RS primarily a function of GWAS size ~ number of loci identified - and with this RS will change with the next bigger GWAS? Also, can the authors speculate (or ideally provide evidence) on other factors beyond number of loci identified that may influence RS for a therapeutic areas (such as genetic architecture underlying the indication or ease of clinical development e.g. as for endocrine and respiratory which have translatable biomarkers)?

4. Related to Comment 3, it is good to see that increased GWAS sample sizes continue to validate

approved drug targets as genetic, but the authors' analysis leaves open whether we are starting to see diminishing returns. For instance, for T2D they report that a recent GWAS (Vujkovic et al., 2020) increased the number of genetic targets from 5 to 7, but the case count of this GWAS was more than 3-fold that of previous ones (e.g. Mahajan et al., 2018). It would be interesting whether this trend holds up when the most recent T2D GWAS is considered which has further doubled case counts (Suzuki et al., 2023, PMID: 37034649). Maybe at this example we DO see a slowing of newly identified genetic targets relative to the number new T2D loci (145 in that study)? Or alternatively omnigenic effects DO start revealing more genetic targets but reduce RS?

5. I am excited about the strong statement that for drugs with multiple approved indications "the odds of advancing to a later stage in the pipeline is 80% higher for indications with genetic support". It would be informative if the authors could condition this analysis on whether the "genetic" indication was the first for the drug to be pursued. If other indications were pursued as part of repositioning efforts they will likely have started with an a priori lower probability of success.

6. The authors write that they did not expect an increased probability for launch for indications with a higher number of associated genes. Can they rule out that this is the result of a "leader->follower effect", i.e. one approved genetic target in an indication (e.g. HMGCR) motivates further research and subsequent approvals of other genetic targets for that indication become much easier (e.g. PCSK9, NPCL1,...)? What other samples are there that may confound the estimated impact of genetics?

7. It is intuitive that RS increases with increased confidence in the causal gene, but assignment of causality in this study remains shallow: they primarily use distance (3 independent SNPs within 1Mb) as evidence for a gene target and for a subset of analyses colocalization, which are at best a proxies. If systematic Mendelian Randomization and colocalization are out of scope for this study, they should at least refer to earlier work where such more thorough causality assessments have further strengthened RS (e.g. Zheng et al., 2020 (<https://www.nature.com/articles/s41588-020-0682-6>) which the authors have contributed to, but here are others). Related to that it'd be interesting to learn whether an "allelic series" further strengthens RS (though I understand that a systematic study of this will probably also go beyond the scope of this study).

8. I applaud the authors for thoroughly documenting tools and sharing data on a GitHub link. While the documentation around targets and indications is intuitive and comprehensive, there is only very little information provided on the actual drugs (apart from T2D). If Pharmaprojects would be willing to make also these data available it would certainly further facilitate broader usage of results from this paper.

9. Based on earlier work the study's main claim that genetic support increases probability of success is by now nearly unambiguous in the field.

Nevertheless, the authors rightfully state that genetics has not yet fully lived up to its expectations and in my opinion it deserves more discussion why this may be the case, not the least to keep expectations to genetics for near term return on investments in check. Explanations include that genetic targets are often harder to drug, more frequently require upregulation, tend to benefit only subsets of a disease population or may not yield commercial success, among others. Particularly the

latter point is now magnified by the authors' new finding that the number of indications for a drug is inversely associated with genetic support (Fig2H, K), which despite higher RS for a lead indication may reduce the appetite to initiate genetic programs.

Other comments:

10. Please specify Column labels Suppl Tables - currently generic and difficult to look at in isolation

11. Fig1b lists FinnGen with an RS of ~ 2.7 , but there are no details provided on the source of these data (although they come reassuringly close to earlier estimates, e.g. in Sun et al, 2022).

12. For Fig2A, please explain what is meant by the therapy area "signs"

13. That RS is independent of OR and effect size is an important message of this study. I'd like to encourage the authors to use the opportunity and explain for a primarily non-geneticist audience why this is unsurprising. If space should be an issue they could consider shortening discussion of the omnigenic model which as they state is not yet limiting.

14. Some of the content in Legend to Figure 2 does not seem to match the respective panel. Please correct.

15. FigS2F: It is worthwhile to mention somewhere that for $\sim 50\%$ of launched targets assessed the association was identified only post launch - supporting the authors' prediction that we will see the fraction (and likely RS) of genetically supported drugs rise further with more genetic data becoming available.

Referee #3 (Remarks to the Author):

I greatly enjoyed this manuscript, which builds on some of the authors 2015 work that was seminal in the area, has been cited more than 1000 times and is widely quoted. The substantial increase in the availability of genotype-phenotype signals since 2015 provides a timely opportunity to update and extend the original work, exploring the impact of several potential modifying factors on the likelihood of drug success. There are a number of important findings, from some that are perhaps unsurprising (e.g. variant effect size doesn't seem to modify RS) to others that seem likely to be valuable and useful (e.g. increasing confidence in the causal gene and increasing phenotypic similarity both improve RS). The manuscript contains a lot of thoughtful and detailed analysis, that is generally well-presented and clear. Nevertheless, I had several general comments and queries (listed below, in no particular order), as well as some more specific comments.

Choice of primary outcome: Given that other non-efficacy factors (e.g. economics, safety concerns) can also have a strong influence on filing for approval for a drug that successfully passes Phase 3, should passing Phase 3 be arguably the primary benchmark here, rather than launch? For example, CETP inhibitors are genetically supported for cardiovascular disease and showed success in Phase 3 in the REVEAL trial, but were not subsequently pursued for approval despite the pathway being validated as a therapeutic target.

Context: I was surprised the authors didn't make much attempt to put their results into context of other literature in the area, such as King et al PLoS Genetics 2019 and Hingorani et al Sci Rep 2019, given that the results appear to be sometimes concordant (e.g. no strong influence of variant effect size on likelihood of success) but sometimes appear to disagree (e.g.

Interpretation of results / statistical power: the paper would benefit from a common approach to interpreting the comparisons made, as there are currently inconsistencies. For example, interpretation of Figure 1b is that OMIM has the highest RS, which is interpreted to be a real finding, whereas for Figure 1d, despite 2007-2010 having a substantially higher RS than other years (~3 vs ~2), this finding is discounted on the basis of a non-significant test for difference. Given the relatively smaller numbers of pairs in some of the subgroups, it would be useful to know what level of statistical power there is in these analyses to be able to put the P-values into context. It would also be useful to give the reader a clear sense of what a meaningful difference in RS is (i.e. if an RS of 2 is important, presumably an RS of 3, or even 2.5, would be importantly higher?). For example, the authors state that "RS is largely unaffected by... year of discovery", but this seems to be potentially untrue according to Figure 1d, which suggests that RS was closer to 3 in 2007-2010 but closer to 2 since.

Robustness of results to choices made: Of course an analysis framework of this sort inherently depends upon the many choices that have to be made. For example, the Hingorani et al paper mentioned above has an extensive set of supplementary material devoted to the importance of each assumption made and the choice of parameters used, much of which is lacking in this manuscript, making it difficult to tell how robust the findings are. As one specific example, for GeneBASS, if I understood correctly, the authors chose to restrict to only those results from the pLoF filter and

the SKAT-O test. But presumably using a broader set of results from Genebase could give somewhat different results?

Robustness of results to data sources used: The results are also, of course, dependent on the data sources used and their strengths and limitations. For example, Genebase is a useful source of gene-phenotype associations from WES analyses, but it only involves results from a single study (UK Biobank). So although this study is large in terms of having >400k participants with WES data, the relative lack of case numbers for common diseases (e.g. type 2 diabetes) means that

Broader uses of human genetics: It is important to note that this analysis only directly addresses one specific use of human genetics in terms of positive support for targets that are ultimately successful. This would be useful to make clearer in the manuscript, to highlight that many people are also using human genetics in other ways to inform drug development, including negative evidence (i.e. deprioritisation of potential therapeutic targets that have strong evidence against a likely causal effect from human genetics) and safety (i.e. identification of potential safety concerns for a therapeutic target from association of relevant genetic variants with other phenotypes). Otherwise I think the paper runs the risk of being potentially misinterpreted to show that the value of human genetics for drug development is somewhat limited. For the same reason, I think it is overinterpreting these results to state that: "human genetic data have not yet begun to appreciably influence pipeline composition across the industry".

Data availability: It was great to see the availability of data sources and code in Github, and some of the Supplementary Tables were useful (although not cited anywhere?) However, I found that as a reader I was missing a supplementary table of the full results that would allow me to look up genes / indications of interest to see their features. Supp Table 5 is very useful in this regard, but seemed to be limited to a subset of gene-indication pairs.

Multiple predictors: The authors tested several lines of genetic evidence individually, but it would potentially be useful to know if the RS is even greater for targets that are supported by multiple lines (e.g. OMIM + OTG). This might provide useful information for readers wishing to use this manuscript as empirical evidence for improving their processes.

Direction of effect: As far as I could tell, the analysis used only considers whether there is a genetic signal for a gene target of interest, but not the direction. Of course, it may be that human genetics provides evidence for an adverse effect of a target on an indication, or even opposing directions for different indications (e.g. as seen for interleukin-6 pathways). Isn't this a problematic flaw with the analysis here, given that evidence in the 'wrong' direction is being counted as genetic support, just as evidence in the 'right' direction?

Specific comments:

- In the 4th paragraph, it is stated that there is an overlap of 2,153 T-I and G-T pairs with an indication-trait similarity ≥ 0.8 , and Fig S2A is cited, but I struggled to see how this figure fits with the text. In Fig S2A, unless I am misunderstanding the plot, it looks to me like setting an indication-trait similarity of 0.8 would give only <1000 genetically supported T-I pairs? (Also there seems to be a small discrepancy with Figure S1, which has 2,152 rather than 2,153)

- The authors mention that OMIM perhaps does better than GWAS due to better causal gene

assignment, so I was surprised that they didn't mention that Genebass (which uses rare coding variants from WES) also appears to be somewhat higher than the various GWAS in Figure 1b, potentially supporting the authors' hypothesis. On the same plot, I also wondered why the Neale UKBB GWAS has higher RS than other GWAS evidence (e.g. GWAS Catalog, FinnGen)? Is this a function of the phenotypes studied perhaps? Or could it be due to the upward bias for the lower frequency variants in the Neale UKBB GWAS that resulted from applying a traditional logistic model using HAIL rather than a logistic mixed model? (As an aside, there seemed to be no methods text on the Neale UKBB GWAS)

- I didn't follow why it was counter to the authors' expectations that there was a higher probability of launch with a greater number of associated genes?

- I wasn't clear why the authors restricted the analyses to GWAS traits with at least 3 independently associated loci – what was the rationale for this implementing this type of filter and for choosing $n=3$, and would it make much difference if alternative values were used?

- I understand the PICCOLO method, but it wasn't clear where the eQTL data were selected from?

- Figure S2C – is there an accidental duplication of “72/436” on the top 2 rows?

Author Rebuttals to Initial Comments:

Referee #1:

Minikel et al are following up to Nelson et al's original seminal work from 2015 that clearly demonstrated the value of human genetics for target discovery. This work and the studies that followed it, certainly under-pinned further investments from pharma companies such as GSK, AZ, Pfizer, Lilly, Regeneron etc in human genetics.

Here Nelson follows up with a more systematic approach to evaluating human genetic support and compare across different therapeutic areas, addressing genetic effect size relationship to therapeutic impact and the importance of genetics by development phase. It's interesting work that I believe will be helpful for the field. I have a few suggestions.

We thank the reviewer for the interest in our work and insightful and constructive comments that follow.

1. The title is very broad but I am not convinced that the paper, is as broad as the title suggests. Fundamentally, most pharma now buy into the relevance of human genetics for research purposes, particularly for target ID. But two important areas that pharma are using human genetics for already that are not well addressed in the current manuscript are safety and Biomarker ID.. Amgen bought DECODE in about 2012 with the primary motive of using genetics to identify potential adverse effects. With the emergence of high throughput molecular data and longitudinal health data, married together with genetics, pharma are also using genetics for Biomarker discovery. While I appreciate that this is a lot of extra effort to fold into the manuscript, and is likely not possible, perhaps the authors could at least comment to these subjects also?

These are expansive topics that would be difficult to fit into the present manuscript given the word limit and format. We have just preprinted a new manuscript as a follow-on to the present work, using a similar approach to show that genetic associations predicted side effects of marketed drugs — that paper is now cited in the 2nd paragraph, along with a new citation to Trajanoska 2023, an excellent Nature review with a broader focus than we can accommodate here. See <https://www.medrxiv.org/content/10.1101/2023.12.12.23299869v1>.

2. The paper would benefit from much more clarity of description. Currently, probably in a bid to reduce word count, it takes some effort to understand. E.g. RS is mentioned lines 17&18, described in lines 44/45 and defined in lines 344, where the elements in the definition are not given e.g. what is X?

We have rewritten the referenced introductory paragraph, as this is supposed to stand alone as an abstract per Nature format, to not introduce the term RS. We have added a new "Definition of metrics" section at the very top of Methods to more thoroughly introduce P(G), P(S), and RS, since these are critical to understanding the whole manuscript. We then deleted redundant description from the Statistics paragraph.

2b. The T2D MVP paper is described but what is meant by novel is not clearly defined is it novel in that a gene has not been characterised before. Is it novel as genetics has not shown an association with T2D before. Is it novel because genetics has not shown an

association with T2D nor glycemic traits?. Is it novel because it's not been associated with T2D, glycemic traits, and is not known biology/pathway?

Here "novel" just meant that the genes had not had a GWAS-based genetic association to T2D before. Rephrased to avoid the word "novel": "When these GWAS quadrupled the number of T2D associated genes from 217 to 862, new genetic support was identified..."

2c. Human gene target is used a lot, but most therapies still target proteins that are encoded by genes. Are the authors restricting to gene targets?

Added a brief parenthetical in the Results text: "assigned a human gene target (usually the gene encoding the target protein)" And in Online Methods, under Drug development pipeline: "Targets were defined as genes; as most drugs do not directly target DNA, this usually refers to the gene encoding the protein target that is bound or modulated by the drug."

2d. Indication-trait similarity threshold is used line 48 but what is it?

Substantially revised this paragraph to provide more stepwise explanation of each concept: "We filtered Citeline Pharmaprojects for monotherapy programs added since 2000 annotated with a highest phase reached and assigned both a human gene target (usually the gene encoding the target protein) and a developed indication defined in Medical Subject Headings (MeSH) ontology. We obtained 29,476 target-indication (T-I) pairs for analysis (Fig. S1). Intersecting with 79,095 unique human gene-trait (G-T) pairs, with traits also defined as MeSH terms, yielded an overlap of 2,153 T-I and G-T pairs (7.3%) where the indication and the trait MeSH terms had Lin/Resnik similarity ≥ 0.8 (Fig. S1, S2A, see Methods)."

2e. Could the authors review for clarity the whole MS.

We have reviewed the whole manuscript and made extensive changes for clarity. These are particularly concentrated in the paragraph introducing our analytical framework, the concluding paragraphs, and the first few sections of Online Methods, but are present throughout. We thank the reviewer for this important suggestion.

3. Stats description should be improved for clarity and justification of significance thresholds provided. Including for PICCOLO - Why so lenient a threshold for eQTL vs GWAS?

Added: "both thresholds determined empirically²⁰ to strongly predict colocalization results."

4. Lack of description of the data sources e.g. the link to the website (ref 18) re-directs to citeline.com and not the website with any browsable data. What is in citeline? Is it all submissions to FDA, or are other medicines authorities also included?

Added a brief description: "Citeline Pharmaprojects¹⁸ is a curated database of drug development programs including preclinical, all clinical phases, and launched drugs." Citeline changed the URL to <https://www.citeline.com/en/products-services/clinical/pharmaprojects> — in case it ever changes yet again, we have now changed the citation to refer to a web.archive.org permalink: <https://web.archive.org/web/20230830135309/https://www.citeline.com/en/products-services/clinical/pharmaprojects>

We have also reviewed our Online Methods section and added similar brief explainers for each other data source as well as for similarity score calculation.

5. I would like the authors to be really crisp on how they define human genetics support. It's not entirely clear to me how this is done. Perhaps a schema could be included as a main figure as this is key to the conclusions drawn? Are all evidences treated equally?

Edited the paragraph introducing the datasets and approach to read: "To characterize the drug development pipeline, we filtered Citeline Pharmaprojects for monotherapy programs added since 2000 annotated with a highest phase reached and assigned both a human gene target (usually the gene encoding the drug target protein) and an indication defined in Medical Subject Headings (MeSH) ontology. This resulted in 29,476 target-indication (T-I) pairs for analysis (Extended Data Fig. 1A). Multiple sources of human genetic associations totaled 81,939 unique gene-trait (G-T) pairs, with traits also mapped to MeSH terms. Intersection these datasets yielded an overlap of 2,166 T-I and G-T pairs (7.3%) where the indication and the trait MeSH terms had a similarity ≥ 0.8 ; we defined these T-I pairs as possessing genetic support (Extended Data Fig. 1B, 2A, see Methods)." We also added a new diagram as Extended Data Figure 1B depicting how genetic support is defined in our database.

Minor comments

Line 30/31 - that GWAS are less predictive of clinical success than Mendelian disease is not surprising given that the causal gene is known for Mendelian disease, while GWAS only points to a region of the genome. But we are not comparing the same thing and there is more to it than this simple comparison. There is considerable investment in human genetics by pharma and decisions are now being made based on human genetics in the pharma setting eg targets being stopped due to lack of human evidence or or potential safety suggested by genetics.

Added: "The differences common and rare disease programs face in regulatory and reimbursement environments⁴ and differing proportions of drug modalities⁹ likely contribute as well." And we have cited Thomas 2021 and Trojanoska 2023 which discuss this difference in greater detail.

Line 342 what are G and S in P(G) P(S)?

We have added parentheticals in the Results text to introduce the abbreviations P(G) and P(S) at the first usage or relevant figure reference, and a "Definition of metrics" paragraph at the top of Methods to describe the metrics in detail. The g and s are underlined to indicate what they stand for: "probability of genetic support, or P(G)" "probability of success, or P(S)".

Some of the databases have been accessed 2-3 years ago eg IntOGen and OMIM. Why not more current? OMIM has been updated. The stats on numbers of gene-phenotype associations look quite different as -provided by OMIM in current release vs quoted in the paper. For others eg open targets, when the database was accessed is completely missing.

We have updated the analysis to the May 31, 2023 release of IntOGen and the September 21, 2023 release of OMIM. Because many of the new OMIM phenotypes initially lacked MeSH mappings, we also went back and updated all of our OMIM-MeSH mappings as well, and re-mapped all Supplementary Concept Records (which often represent interim MeSH terms for

concepts not yet incorporated into the MeSH hierarchy) to MeSH main headings. The updating of these datasets, mappings, and recomputation of our entire similarity matrix have resulted in minor numerical changes to results throughout.

The omni-genie model discussion is a bit of a distraction has doesn't add to the story. I think all the authors are trying to say is that it's difficult to go from list of 100s of causal genes to a short prioritised list of potential targets, which is not controversial.

We have substantially shortened the discussion of the omnigenic model.

For locus to gene share - is this defined per disease or is it across diseases? Can this be clarified line 268.

Added: "OTG provides an L2G score for the triplet of each study or trait with each hit and each possible causal gene." For the reviewer's benefit, we note that while OTG defines L2G per disease/trait, the scores generally do not vary much for the same SNP across different studies.. For instance, 20-004699605-A-G is assigned 86% to PRNP for study GCST000294, vs. 88% to PRNP for study GCST90001389.

The authors might like to comment on the lag time between genetic discovery and clinical translation. This is an important feature as the genetic targets that are identified today wont reach first human dose for likely 5-10 years. Therefore we will see a continued growth in targets that have genetic support given the large continuing investment in genetics cohorts by pharma eg ukb, finngen

Added: "since drug programs prospectively supported by human genetics have had a mean lag time from genetic association of 13 years to first trial²⁰ and 21 years to approval⁹". Both of the papers now cited here used manual curation to determine date of discovery for Mendelian associations (which is not annotated in OMIM) and so provide a better estimate of lag time than our own data do.

Referee #2:

In this manuscript, the authors convincingly demonstrate that drugs directed against targets with human genetic support have a higher probability of regulatory approval. Using the latest genetic datasets they refine earlier probability of success estimates for a range of therapeutic areas, explore which genetic mechanisms are more versus less likely to differentiate successful from unsuccessful drugs under development and conclude that thus far only a minority of genetically supported target-indication pairs have been leveraged for drug discovery.

While the conclusions from this study are not fundamentally novel, this is an important and timely piece of work that builds upon and expands earlier studies exploring the potential of human genetics to reduce the horrendous failure rates in drug development. The study is elegantly designed, technically straightforward and the statistics and conclusions seem appropriate.

We thank the reviewer for this helpful critical read of our work and for the interest in our study.

1. In their seminal 2015 paper the authors provided evidence that the fraction of genetic targets increases with pipeline stage (i.e., approved > Ph III > Ph II > Ph I). Based on their new analysis it now does no longer seem to hold up that genetic support in general reduces the risk of failure to progress to the next development phase, with modest separation starting only between Ph II and Ph III and a substantial separation not before Ph III and Launched (Fig1A). The earlier observation though still seems to hold up for drugs with one target only (FigS6) as well as for distinct therapeutic areas (Fig2A-F). If this interpretation is correct, this refinement should be stated and discussed in the manuscript.

We have added a new Extended Data Figure 5 showing P(G) by phase and by therapy area. It appears that some therapy areas do have a stepwise increase across clinical phases, while in others the jump occurs entirely at launch. Added text: "Accordingly, therapy areas differed in P(G) and in whether P(G) increased throughout clinical development or only at launch (Extended Data Fig. 5)."

2. Notably, P(G) appears higher for all categories and phases in the current paper than in the authors' previous paper. For instance, in 2015 P(G) (if equivalent) for Ph I was ~2% for OMIM+GWASdb while now it is ~4%, even for the historical category. Is this primarily a reflection that thanks to growing genetic datasets we can now just find more genetic support for existing Development targets? Or does it also come from an increased number of new genetic entries into the Development space? (Or could it simply be because the definition of genetic support is slightly different between the two papers?) More granularity on which factors may have doubled P(G) since 2015, and in which phases this effect is most prominent, would be appreciated.

Added: "In each phase, P(G) was higher than previously reported^{5,14}, owing, as expected^{14,15}, more to new G-T discoveries than to changes in drug pipeline composition (Extended Data Fig. 3A-F)." The newly added Extended Data Figure 3 shows P(G) at each phase for every combination of current genetic and drug pipeline datasets versus Nelson 2015 genetic and drug datasets, plus a breakdown of the current drug dataset with pre-2013 vs all-time OTG GWAS hits. Overall, the largest differences in P(G) are between old and new genetic datasets, with minimal change when the old vs. new drug pipeline data are used.

3. In the authors' 2015 analysis, "musculoskeletal" had been the therapeutic area with the highest enrichment for genetic targets among approvals. In the current paper this therapeutic area shows the lowest RS across all phases. Conversely, "metabolic" stays near the top in both analyses. This still seems to be the case after re-analysis of the 2015 data for RS (FigS4F). What explains such disparities? Is it only because the "search space" (i.e. available genetic data) has been considerably growing in some indications (e.g. metabolic; Fig2F) while for others (e.g. musculoskeletal) not so much. In other words: Is RS primarily a function of GWAS size ~ number of loci identified - and with this RS will change with the next bigger GWAS? Also, can the authors speculate (or ideally provide evidence) on other factors beyond number of loci identified that may influence RS for a therapeutic areas (such as genetic architecture underlying the indication or ease of clinical development e.g. as for endocrine and respiratory which have translatable biomarkers)?

We have added a new panel, Extended Figure 7G, showing a confusion matrix of how drug indications from Nelson 2015 were categorized in Nelson 2015 versus in our present work. For

musculoskeletal in particular, we note that the number of genetically supported T-I pairs in 2015 was small (N=11); the confusion matrix also shows that while N=25 unique drug indications were classified as musculoskeletal both in 2015 and in 2023, more than half of the indications now classified as "musculoskeletal" were either absent from the 2015 dataset (N=24) or were classified differently (N=10). We have also added Extended Data Figure 5 showing how the proportion of programs with genetic support, P(G), varies at each phase across the therapy areas considered today. The question about whether translatable biomarkers contribute to differences in RS is an interesting one but beyond the scope of the present work, we have added references to Trajanoska 2023 and Burgess 2023, both of which provide a broader review of the topic than we have space to do in this manuscript.

4. Related to Comment 3, it is good to see that increased GWAS sample sizes continue to validate approved drug targets as genetic, but the authors' analysis leaves open whether we are starting to see diminishing returns. For instance, for T2D they report that a recent GWAS (Vujkovic et al., 2020) increased the number of genetic targets from 5 to 7, but the case count of this GWAS was more than 3-fold that of previous ones (e.g. Mahajan et al., 2018). It would be interesting whether this trend holds up when the most recent T2D GWAS is considered which has further doubled case counts (Suzuki et al., 2023, PMID: 37034649). Maybe at this example we DO see a slowing of newly identified genetic targets relative to the number new T2D loci (145 in that study)? Or alternatively omnigenic effects DO start revealing more genetic targets but reduce RS?

We have updated the analysis to incorporate new hits from both Vujkovic 2020 and Suzuki 2023 as novel GWAS. Note that Suzuki 2023 has not yet been ingested by Open Targets Genetics, and the genes listed for each locus in Suzuki 2023 Supplementary Table 4 appear to be assigned based simply on proximity (at least, no more sophisticated gene assignment approach is described in their Methods). Therefore, to map the Suzuki 2023 loci to genes, we used OTG L2G scores that had already been assigned for the same lead SNPs in other OTG GWAS; this yielded a top gene with L2G share > 0.5 for 27% of the hits. This methodology limitation is now updated in our Methods.

(Aside: note that this update also slightly alters the denominators in the supplementary figure — this is because, as described in Methods, for drugs with multiple targets, if one target has genetic support then we use that target. For instance, Phase I drug candidate MK-3655 was listed as targeting both FGFR1 and KLB; in the previous version, this drug was represented by FGFR1 as target, but now that KLB has genetic support from Suzuki 2023, this drug is represented by KLB, which had already appeared in Phase I for drug candidate BFKB-8488A. Thus, these two drugs collectively contribute one fewer unique target at Phase I. Examples like this one caused the count of T2D targets in Phase I to drop from 39 to 36.)

Overall, the effect of adding in the Suzuki data is that 1) long-known OMIM targets (GCK and INSR) now have GWAS support for the first time, so the "Launched" category in panel C changes from 2/12 to 4/12, and 2) the number of supported programs in Preclinical, Phase I, and Phase II in panels C and D increases slightly. There remains a very large gap between P(G) at any clinical phase and P(G) for Launched, consistent with the original message of this figure, which is that newer, larger, GWAS still meaningfully differentiate clinical-stage programs from launched programs.

At the same time, we also considered the reviewer's comment about "diminishing returns" — adding only 2 new launched targets when the number of GWAS loci increases by hundreds — and decided it would be fruitful to also include a discussion of "yield" — i.e. the proportion of

genetically supported gene-disease links that actually represent approved drugs. We therefore added the following paragraph near the end of the paper:

"The focus of this work has been on the relative success of drug programs with and without genetic evidence, limited to drug mechanisms that have entered clinical development. This metric does not address the probability that a gene associated with a disease, if targeted, will yield a successful drug. At the early stage of target selection, is evidence of a large loss of function effect in one gene usually a better choice than a small non-coding SNP effect on the same phenotype in another? We explored this question for T2D studies referenced above. Of the 7 targets of launched drugs with genetic evidence, 4 had Mendelian evidence (in addition to pre-2020 GWAS evidence), out of a total of 19 Mendelian genes related to T2D (21%). 1 launched T2D target had only GWAS (and no Mendelian) evidence among 217 GWAS associated genes prior to 2020 (0.46%), while 2 launched targets were among 645 new GWAS associations since 2020 (0.31%). At least in this example, the "yield" of genetic evidence for successful drug mechanisms was greatest for genes with Mendelian effects, but similar between earlier and later GWAS. Clearly, just because genetic associations differentiate clinical stage drug targets from launched ones does not mean that a large fraction of associations will be fruitful. Moreover, genetically supported targets may be more likely to require upregulation or be druggable only by more challenging modalities^{4,9}. More work is required to better understand the challenges of target identification and prioritization given the genetic evidence precondition."

5. I am excited about the strong statement that for drugs with multiple approved indications "the odds of advancing to a later stage in the pipeline is 80% higher for indications with genetic support". It would be informative if the authors could condition this analysis on whether the "genetic" indication was the first for the drug to be pursued. If other indications were pursued as part of repositioning efforts they will likely have started with an a priori lower probability of success.

Unfortunately, Pharmaprojects captures phase progression dates only on a per-drug basis (i.e. the first year that each drug reached Phase I, II, III, and Launch for any indication), along with highest phase reached for each drug-indication pair (i.e. the highest phase ever reached for each indication, regardless of year). Pharmaprojects does not have indication-level progression dates, thus, we do not systematically know which indication was first.

6. The authors write that they did not expect an increased probability for launch for indications with a higher number of associated genes. Can they rule out that this is the result of a "leader->follower effect", i.e. one approved genetic target in an indication (e.g. HMGCR) motivates further research and subsequent approvals of other genetic targets for that indication become much easier (e.g. PCSK9, NPCL1,...)? What other samples are there that may confound the estimated impact of genetics?

Added: "This is unlikely due to successful genetically-supported programs inspiring other programs, as most genetic support was discovered retrospectively (Extended Data Fig. 2G); the few examples of drug programs prospectively motivated by genetic evidence were primarily for Mendelian diseases⁹." (The citation is to Trajanoska 2023).

7. It is intuitive that RS increases with increased confidence in the causal gene, but assignment of causality in this study remains shallow: they primarily use distance (3 independent SNPs within 1Mb) as evidence for a gene target and for a subset of analyses colocalization, which are at best a proxies. If systematic Mendelian Randomization and colocalization are out of scope for this study, they should at least refer to earlier work

were such more thorough causality assessments have further strengthened RS (e.g. Zheng et al., 2020 (<https://www.nature.com/articles/s41588-020-0682-6>) which the authors have contributed to, but here are others). Related to that it'd be interesting to learn whether an "allelic series" further strengthens RS (though I understand that a systematic study of this will probably also go beyond the scope of this study).

First, we clarified in Methods > Drug development pipeline that our criteria for deciding that a trait has been studied genetically ("genetic insight", which includes the 1 Mb distance cutoff) are distinct from the criteria used for gene mapping:

"Indications were considered to possess "genetic insight" — meaning the human genetics of this trait or similar traits have been successfully studied — if they had ≥ 0.8 similarity to i) an OMIM or IntOGen disease, or ii) a GWAS trait with at least 3 independently associated loci, based on lead SNP positions rounded to the nearest 1 Mb. For calculating relative success, we used the number of T-I pairs with genetic insight as the denominator. The rationale for this choice is to focus on indications where there exists the opportunity for human genetic evidence, consistent with the filter applied previously⁵. However, we observe that our findings are not especially sensitive to the presence of this filter, with RS decreasing by just 0.17 when the filter is removed (Extended Data Fig. 3G-H). Note that the criteria for determining "genetic insight" are distinct from, and much looser than, the criteria for mapping GWAS hits to genes (see locus-to-gene or L2G scores under Open Targets Genetics below)."

Next, we clarified the basis for the L2G scores provided by Open Targets Genetics. This method was published by Mountjoy et al 2021 <https://www.nature.com/articles/s41588-021-00945-5> and is cited in our paper, but we briefly review it here while also noting the limitation that we are not utilizing more sophisticated colocalization or MR approaches, citing Zheng 2020 as an example:

"OTG L2G scores used for gene mapping are based on a machine learning model trained on gold standard causal genes²⁷; inputs to that model include distance, functional annotations, eQTLs, and chromatin interactions. Note that we do not utilize Mendelian randomization²⁸ to map causal genes, and even gene mappings with high L2G scores are necessarily imperfect."

Finally, related to the "allelic series" comment, we now show in Extended Data Fig. 2B that T-I pairs with both OMIM and GWAS support have the highest RS of all, 4.1.

8. I applaud the authors for thoroughly documenting tools and sharing data on a GitHub link. While the documentation around targets and indications is intuitive and comprehensive, there is only very little information provided on the actual drugs (apart from T2D). If Pharmaprojects would be willing to make also these data available it would certainly further facilitate broader usage of results from this paper.

Pharmaprojects granted us permission to release an open dataset derived from their commercial database only at the level of target-indication pairs. Unfortunately we do not have permission to release drug-level information.

9. Based on earlier work the study's main claim that genetic support increases probability of success is by now nearly unambiguous in the field. Nevertheless, the authors rightfully state that genetics has not yet fully lived up to its expectations and in my opinion it deserves more discussion why this may be the case, not the least to keep expectations to genetics for near term return on investments in check. Explanations include that genetic targets are often harder to drug, more

frequently require upregulation, tend to benefit only subsets of a disease population or may not yield commercial success, among others. Particularly the latter point is now magnified by the authors' new finding that the number of indications for a drug is inversely associated with genetic support (Fig2H, K), which despite higher RS for a lead indication may reduce the appetite to initiate genetic programs.

Added: "Moreover, genetically supported targets may be more likely to require upregulation, to be druggable only by more challenging modalities^{4,9}, or to enjoy narrower use across indications."

Other comments:

10. Please specify Column labels Suppl Tables - currently generic and difficult to look at in isolation

We thank the reviewer for this suggestion. We have added more descriptive column names to every supplementary table, and have additionally provided a table of abbreviations within the supplement to improve readability.

11. Fig1b lists FinnGen with an RS of ~2.7, but there are no details provided on the source of these data (although they come reassuringly close to earlier estimates, e.g. in Sun et al, 2022).

Added in the figure legend: "OTG links were parsed to determine the source of each OTG data point: the EBI GWAS catalog²⁹ (N=136,503 hits with L2G share ≥ 0.5), Neale UK BioBank (<http://www.nealelab.is/uk-biobank>; N=19,139), FinnGen R6³⁰ (N=2,338), or SAIGE (N=1,229)."

The reviewer's comment also led us to add a clarification in Methods citing the Sun et al paper: "For clarity, we note that where other recent studies^{15,18} have examined the fold enrichment and overlap between genes with a human genetic association and genes encoding a drug target, without regards to similarity, herein all of our analyses are conditioned on the similarity between the drug's indication and the genetically associated trait."

Specifically, the relevant section from Sun et al suggests they conducted an analysis only of targets (T) and not target-indication (T-I) pairs: "We found a statistically significant enrichment of significant genes in our study that were also approved drug targets (26 out of 482, compared with a background of 569 approved targets out of 19,955 genes, OR=1.9, P=0.0024), which is in line with previous estimates of a higher success rate for drug targets supported by genetics^{22,23}. Sensitivity analyses using more stringent association P-value thresholds further increased these probability estimates (P=5 × 10⁻⁹ (OR=2.3, P=0.00070); P=5 × 10⁻¹⁰ (OR=2.5, P=0.00037))"

12. For Fig2A, please explain what is meant by the therapy area "signs"

Changed to "signs/symptoms" and added explanation in Methods: "MeSH terms for Pharmaprojects indications were mapped onto 16 top-level headings under the Diseases [C] and Psychiatry and Psychology [F] branches of the MeSH tree (<https://meshb.nlm.nih.gov/treeView>), plus an "other". The signs/symptoms area corresponds to C23 Pathological Conditions, Signs, and Symptoms and contains entries such as inflammation and pain."

13. That RS is independent of OR and effect size is an important message of this study. I'd like to encourage the authors to use the opportunity and explain for a primarily non-

geneticist audience why this is unsurprising. If space should be an issue they could consider shortening discussion of the omnigenic model which as they state is not yet limiting.

Added: "One reason for this is likely because genetic effect size on a phenotype rarely accounts for the magnitude of genetic effect on gene expression, protein function, or some other molecular intermediate. In some circumstances, genetic effect sizes can yield insights into anticipated drug effects. This is best illustrated for cardiovascular disease therapies, where genetic effects on cholesterol and disease risk and treatment outcomes are correlated²²." We correspondingly removed substantial discussion regarding the omnigenic model.

14. Some of the content in Legend to Figure 2 does not seem to match the respective panel. Please correct.

We have fixed the legend, apologies for that error.

15. FigS2F: It is worthwhile to mention somewhere that for ~50% of launched targets assessed the association was identified only post launch - supporting the authors' prediction that we will see the fraction (and likely RS) of genetically supported drugs rise further with more genetic data becoming available.

Added in main text: "most genetic support was discovered retrospectively (Extended Data Fig. 2G)". In response to other reviewer comments, we also added a new Extended Data Figure 3 showing how P(G) and RS change using 2013 vs. 2023 datasets, which also addresses this point. In the legend, we have added as suggested the following statement: "We note that both the contrasts in this figure, and the fact that genetic support is so often retrospective (Figure Extended Data Fig. 2G) suggest that P(G) will continue to rise in coming years."

Referee #3

I greatly enjoyed this manuscript, which builds on some of the authors 2015 work that was seminal in the area, has been cited more than 1000 times and is widely quoted. The substantial increase in the availability of genotype-phenotype signals since 2015 provides a timely opportunity to update and extend the original work, exploring the impact of several potential modifying factors on the likelihood of drug success. There are a number of important findings, from some that are perhaps unsurprising (e.g. variant effect size doesn't seem to modify RS) to others that seem likely to be valuable and useful (e.g. increasing confidence in the causal gene and increasing phenotypic similarity both improve RS). The manuscript contains a lot of thoughtful and detailed analysis, that is generally well-presented and clear. Nevertheless, I had several general comments and queries (listed below, in no particular order), as well as some more specific comments.

Choice of primary outcome: Given that other non-efficacy factors (e.g. economics, safety concerns) can also have a strong influence on filing for approval for a drug that successfully passes Phase 3, should passing Phase 3 be arguably the primary

benchmark here, rather than launch? For example, CETP inhibitors are genetically supported for cardiovascular disease and showed success in Phase 3 in the REVEAL trial, but were not subsequently pursued for approval despite the pathway being validated as a therapeutic target.

Unfortunately, trial outcomes per se are not recorded in Pharmaprojects. The development phases in the source database are phase I, II, III, registration (meaning an NDA or equivalent has been filed), and launched. For an example like CETP where the sponsor never chose to file an NDA, there is no systematic data captured to indicate that the phase III trial did in fact meet its primary endpoint. Our only way of knowing whether a drug "succeeded" at each phase is whether it reached a subsequent phase or milestone.

Context: I was surprised the authors didn't make much attempt to put their results into context of other literature in the area, such as King et al PLoS Genetics 2019 and Hingorani et al Sci Rep 2019, given that the results appear to be sometimes concordant (e.g. no strong influence of variant effect size on likelihood of success) but sometimes appear to disagree (e.g. " GWAS has similar strength prediction to OMIM with high-confidence causal gene assignment through missense/pLoF in King et al, but OTG has considerably lower RS than OMIM irrespective of L2G threshold in this manuscript in Figure 1C).

We have added several statements comparing or contrasting our results with those noted:

"In each phase, P(G) was higher than previously reported^{5,14}, owing, as expected^{14,15}, more to new G-T discoveries than to changes in drug pipeline composition (Extended Data Fig. 3A-F)"

"We note that the increase in P(G) over the past decade⁵ is almost entirely attributable to new genetic evidence (e.g. contrast B vs. A, D vs. C, F vs. E) rather than changes in the drug pipeline (e.g. compare A vs. C, B vs. D). In contrast, the increase in RS is due mostly to changes in the drug pipeline (compare C, D, E, F vs. A, B), in line with theoretical expectations outlined by Hingorani et al¹⁵ and consistent with the findings of King et al¹⁴."

"RS was highest for OMIM (RS = 3.7), in agreement with prior reports^{5,14}"

"The relatively minor impact of removing the genetic insight filter is consistent with the findings of King et al¹³, who varied the minimum number of genetic associations required for an indication to be included, and found that risk ratio for progression (i.e. RS) was slightly diminished when the threshold was reduced."

"Lack of impact of GWAS Catalog lead SNP odds ratio (OR) on RS when using the same OR breaks as used by King et al¹³"

Interpretation of results / statistical power: the paper would benefit from a common approach to interpreting the comparisons made, as there are currently inconsistencies. For example, interpretation of Figure 1b is that OMIM has the highest RS, which is interpreted to be a real finding, whereas for Figure 1d, despite 2007-2010 having a substantially higher RS than other years (~3 vs ~2), this finding is discounted on the basis of a non-significant test for difference. Given the relatively smaller numbers of pairs in the some of the subgroups, it would be useful to know what level of statistical power there is in these analyses to be able to put the P-values into context. It would also be useful to give the reader a clear sense of what a meaningful difference in RS is (i.e. if

an RS of 2 is important, presumably an RS of 3, or even 2.5, would be importantly higher?). For example, the authors state that “RS is largely unaffected by... year of discovery”, but this seems to be potentially untrue according to Figure 1d, which suggests that RS was closer to 3 in 2007-2010 but closer to 2 since.

We have added Extended Data Figure 6 showing that the categories in panel 1D are each significantly confounded with therapy area. Because therapy areas do have substantial and statistically significant differences in RS, this confounding is one reason we prefer to more cautiously interpret small, non-significant differences in panel 1D. Also note that panel 1D is necessarily a somewhat arbitrary (every 4 years) binning of the data; Extended Data Figure 2F shows the proportion of GWAS Catalog-supported TI pairs that are launched by year; the apparently higher RS in the first bin is driven mostly by 10/20 pairs with 2008 genetic support being launched; overall there does not appear to be any trend from 2007 through 2022.

Robustness of results to choices made: Of course an analysis framework of this sort inherently depends upon the many choices that have to be made. For example, the Hingorani et al paper mentioned above has an extensive set of supplementary material devoted to the importance of each assumption made and the choice of parameters used, much of which is lacking in this manuscript, making it difficult to tell how robust the findings are. As one specific example, for GeneBASS, if I understood correctly, the authors chose to restricted to only those results from the pLoF filter and the SKAT-O test. But presumably using a broader set of results from Genebass could give somewhat different results?

We have added new sensitivity analyses throughout the paper, including:

- We now include four different Genebass queries (missense and LoF, SKAT and burden) in all the main analyses; we then show the breakdown of RS between these different queries in Extended Data Figure 2B.
- Comparison of RS for OMIM and GWAS both without each other, and in combination, also in Extended Data Figure 2B.
- Effect of old vs. new genetic and drug pipeline datasets in Extended Data Figure 3A-F.
- Effect of removing the "genetic insight" filter in Extended Data Figure 3G-H.

We have also highlighted in the text other similarity analyses performed here:

- Varying minimum L2G score for OTG GWAS hist inclusion (Figure 1C)
- Varying similarity threshold (Figure S2A)
- Source of genetic evidence (Figure 1B, Extended Data Figure 2B-C and 6)
- Restriction to drugs with exactly one target (Figures S1-S10)

Robustness of results to data sources used: The results are also, of course, dependent on the data sources used and their strengths and limitations. For example, Genebass is a useful source of gene-phenotype associations from WES analyses, but it only involves results from a single study (UK Biobank). So although this study is large in terms of having >400k participants with WES data, the relative lack of case numbers for common diseases (e.g. type 2 diabetes) means that relatively few genes are identified and therefore the numbers on which to base the RS estimates are lower than they would be if one were to take the largest (i.e. most case-rich) WES for some of these diseases.

Added to discussion: "A limitation is that, other than Genebass, we did not include whole exome or whole genome sequencing association studies, which may be more likely to pinpoint causal variants or directions of effect (such as loss-of-function)."

Broader uses of human genetics: It is important to note that this analysis only directly addresses one specific use of human genetics in terms of positive support for targets that are ultimately successful. This would be useful to make clearer in the manuscript, to highlight that many people are also using human genetics in other ways to inform drug development, including negative evidence (i.e. deprioritisation of potential therapeutic targets that have strong evidence against a likely causal effect from human genetics) and safety (i.e. identification of potential safety concerns for a therapeutic target from association of relevant genetic variants with other phenotypes). Otherwise I think the paper runs the risk of being potentially misinterpreted to show that the value of human genetics for drug development is somewhat limited. For the same reason, I think it is overinterpreting these results to state that: “human genetic data have not yet begun to appreciably influence pipeline composition across the industry”.

We thank the reader for urging us to be more cautious in our interpretation. We have deleted that sentence and replaced it with a milder one: "Instead, we find that active programs possess genetic support only slightly more often than historical programs and remain less enriched for genetic support than approved drugs." While we do not have space to adequately address all of these issues — such as deprioritization and safety — within this manuscript, we have added citations to several papers with a broader focus including Trajanoska 2023 (broader review of prospective use of human genetics in drug discovery); Carss 2023, Nguyen 2019 (safety); and Diogo 2018 (deprioritization).

Data availability: It was great to see the availability of data sources and code in Github, and some of the Supplementary Tables were useful (although not cited anywhere?) However, I found that as a reader I was missing a supplementary table of the full results that would allow me to look up genes / indications of interest to see their features. Supp Table 5 is very useful in this regard, but seemed to be limited to a subset of gene-indication pairs.

We have added this as supplementary table S1 as requested.

Multiple predictors: The authors tested several lines of genetic evidence individually, but it would potentially be useful to know if the RS is even greater for targets that are supported by multiple lines (e.g. OMIM + OTG). This might provide useful information for readers wishing to use this manuscript as empirical evidence for improving their processes.

We have added a breakdown of OMIM and GWAS both alone and in combination in Extended Data Figure 2B. The reviewer was correct to suspect that combined support from both OMIM and GWAS is the strongest of all (RS = 4.1).

Direction of effect: As far as I could tell, the analysis used only considers whether there is a genetic signal for a gene target of interest, but not the direction. Of course, it may be that human genetics provides evidence for an adverse effect of a target on an indication, or even opposing directions for different indications (e.g. as seen for interleukin-6 pathways). Isn't this a problematic flaw with the analysis here, given that evidence in the 'wrong' direction is being counted as genetic support, just as evidence in the 'right' direction?

Agreed, this is an important limitation. Early on in the exploratory analyses that led to this paper, we determined that performing the analysis in a manner aware of direction of effect would be infeasible. The effect on gene function for GWAS hits is not systematically annotated, and indeed, in many cases (e.g. where there is not clear colocalization with an eQTL), it is impossible to know without extensive functional studies; for GWAS hits of quantitative traits, the mapping of those quantitative traits to directional effect on disease risks is also not systematically annotated; assignment of all OMIM genes to gain-of-function versus loss-of-function has not been published and in a subset of cases remains genuinely controversial; perhaps only for IntOGen tumor suppressors vs. oncogenes and for Genebase loss-of-function burden tests is direction of effect generally clear. Adding to these problems, Pharmaprojects annotations as to mechanism of drug action (e.g. agonist versus antagonist) are incomplete. Therefore we have added in the discussion: "Moreover, all of our analyses are naïve to direction of genetic effect (gain versus loss of gene function) as this is unknown or unannotated in most datasets utilized here."

One could speculate that the observed RS would be even larger if we were able to perform these analyses aware of direction of effect. However, it is also possible that pharmaceutical companies are only rarely wrong about the direction of effect desired for their drug, and therefore that the set of programs reaching Phase I is invisibly filtered for correct alignment between drug mechanism and genetic effect, in which case the RS may not be so different from what we observe here. Given that it's impossible to know, we opted not to speculate on this further in our paper.

Specific comments:

- In the 4th paragraph, it is stated that there is an overlap of 2,153 T-I and G-T pairs with an indication-trait similarity ≥ 0.8 , and Fig S2A is cited, but I struggled to see how this figure fits with the text. In Fig S2A, unless I am misunderstanding the plot, it looks to me like setting an indication-trait similarity of 0.8 would give only <1000 genetically supported T-I pairs? (Also there seems to be a small discrepancy with Figure S1, which has 2,152 rather than 2,153)

Thank you to the reviewer for pointing out this discrepancy. Most figures, including Extended Data Figure 2A, use only germline associations, calculate RS from Phase I to Launch (i.e. excluding Preclinical), and use only OTG genes with L2G share ≥ 0.5 . Given those filters, the count is 749 (formerly 743) supported T-I pairs. In contrast the 2,166 figure (formerly 2,153) includes IntOGen, Preclinical, and OTG hits with L2G share < 0.5 . This latter figure is still worth mentioning as it defines the full universe of data considered here in panels that do examine specifically IntOGen (Extended Data Figure 2C), Preclinical (Figure 1A, 2A), or that vary the L2G threshold (Figure 1C). We have updated Extended Data Figure 1A to include both statistics for clarity.

- The authors mention that OMIM perhaps does better than GWAS due to better causal gene assignment, so I was surprised that they didn't mention that Genebase (which uses rare coding variants from WES) also appears to be somewhat higher than the various GWAS in Figure 1b, potentially supporting the authors' hypothesis. On the same plot, I also wondered why the Neale UKBB GWAS has higher RS than other GWAS evidence (e.g. GWAS Catalog, FinnGen)? Is this a function of the phenotypes studied perhaps? Or could it be due to the upward bias for the lower frequency variants in the Neale UKBB GWAS that resulted from applying a traditional logistic model using HAIL rather than a

logistic mixed model? (As an aside, there seemed to be no methods text on the Neale UKBB GWAS)

Apologies for the lack of methods detail — we have added the following: "OTG links were parsed to determine the source of each OTG data point: the EBI GWAS catalog²⁷ (N=136,503 hits with L2G share ≥ 0.5), Neale UK BioBank (<http://www.nealelab.is/uk-biobank>; N=19,139), FinnGen R628 (N=2,338), or SAIGE (N=1,229)." In addition, we have added a new Extended Data Figure 6 showing that therapy areas are distributed non-randomly between the sources of GWAS evidence — in particular, the therapy areas with the highest RS have slightly greater representation among Neale UKBB-supported T-I pairs compared to GWAS Catalog or FinnGen.

- I didn't follow why it was counter to the authors' expectations that there was a higher probability of launch with a greater number of associated genes?

We have removed the phrase "counter to our expectations" and changed this sentence to read "Success rate nominally increased..."

Just to explain for the reviewer our original thought process: the original framing of this section of the paper discussed the omnigenic model, which would hypothesize that as GWAS sample sizes grow, eventually all/most genes will be associated to all/most polygenic traits and therefore, one might suppose that larger GWAS will do more poorly at differentiating between successful and unsuccessful drug mechanisms. However, in response to feedback from reviewers 1 and 2, we are reducing the focus given to the omnigenic model here, in which case saying "counter to our expectations" does not really fit.

- I wasn't clear why the authors restricted the analyses to GWAS traits with at least 3 independently associated loci – what was the rationale for this implementing this type of filter and for choosing n=3, and would it make much difference if alternative values were used?

We have added Extended Data Figure 3G-H showing that the removal of this "genetic insight" filter altogether makes little difference — RS overall drops 0.17, and this varies only a little bit between genetic association sources and between therapy areas. We originally included the filter for consistency with Nelson 2015. Added in Methods: "The rationale for this choice is to focus on indications where there exists the opportunity for human genetic evidence, consistent with the filter applied previously⁵. However, we observe that our findings are not especially sensitive to the presence of this filter, with RS decreasing just 0.17 when the filter is removed (Extended Data Figure 3G-H)."

- I understand the PICCOLO method, but it wasn't clear where the eQTL data were selected from?

Added to Methods paragraph about PICCOLO: "As described²⁹, gene mapping utilizes QTL data from GTEx (N=7,162) and a variety of other published sources (N=6,552)."

- Figure S2C – is there an accidental duplication of “72/436” on the top 2 rows?

We have confirmed that this is not a duplication; because all IntOGen-supported T-I pairs are in oncology, the approved/supported numbers for "All indications" and "oncology" are indeed

identical. We have added a few explanatory sentences in the figure legend to avoid other readers having the same doubt:

"Note that the approved/supported proportions displayed for the top two rows are identical because all IntOGen genetic support is for oncology indications, yet the RS is different because the number of non-supported approved and non-supported clinical stage programs is different. In other words, in the "All indications" row, there is a Simpson's paradox that diminishes the apparent RS of IntOGen — IntOGen support improves success rate (see 2nd row) but also selects for oncology, an area with low baseline success rate (as shown in Extended Data Fig. 6A)."

We have also included a supplementary table (S13) with all of the data behind that figure including the number of unsupported programs that are and are not launched, so that interested readers can reconstruct all cells of the 2x2 contingency table that leads to the RS calculation.

Reviewer Reports on the First Revision:

Referees' comments:

Referee #1 (Remarks to the Author):

Thank you to the authors for addressing my comments. I am happy with the responses. I think the work will be helpful to the field with the most thorough investigation of genetics in drug discovery/development to date.

Referee #1 (Remarks on code availability):

I dont think i am best placed to test the code.

Referee #2 (Remarks to the Author):

My comments have been well addressed. I much appreciate the additional thorough analyses and particularly some differentiated discussion added to the revised version of the manuscript. My remaining comments remain very minor. I congratulate the authors to another great piece of work that should be informative to a broad audience.

Re. My Comment 3: Thank you for outlining why RS for different therapeutic areas now makes them rank differently relative to 2015, as well as adding Extended Data Figure 7. The marginals in the new confusion graph are missing an explanation where they are derived from. I assume the horizontal row is the number of indications that are included in the current analysis, but had been missing in 2015, and the vertical column is the inverse? If so (or other), please add explanation to the Legend or graph.

The sentence introduced in response to my Comment 4 "When these GWAS quadrupled the number of T2D associated hit genes..." currently stands a bit isolated (and can probably be further simplified). It'd be good to refer or link these results to the sound new discussion on the diminishing returns topic further below in the manuscript.

Heiko Runz

Referee #2 (Remarks on code availability):

No, but I confirm it is accessible together with data under the provided link

Referee #3 (Remarks to the Author):

The authors have clearly and thoroughly addressed the comments I provided (and most of those from the other reviewers) and I believe the manuscript is improved as a result. In particular, the addition of ST1 and the definitions/abbreviations in the Supp Tables and main text should make the manuscript easier for readers to interpret and recreate results. While there are some analyses that would have been ideal to be able to incorporate, I understand that the available data is often limited and hence the authors have appropriately listed these as limitations in the text instead.

Author Rebuttals to First Revision:

Referees' comments:

Referee #1 (Remarks to the Author):

Thank you to the authors for addressing my comments. I am happy with the responses. I think the work will be helpful to the field with the most thorough investigation of genetics in drug discovery/development to date.

Referee #1 (Remarks on code availability):

I dont think i am best placed to test the code.

We thank Reviewer 1 for their helpful contribution to improving this manuscript.

Referee #2 (Remarks to the Author):

My comments have been well addressed. I much appreciate the additional thorough analyses and particularly some differentiated discussion added to the revised version of the manuscript. My remaining comments remain very minor. I congratulate the authors to another great piece of work that should be informative to a broad audience.

We thank Reviewer 2 for their incisive constructive comments.

Re. My Comment 3: Thank you for outlining why RS for different therapeutic areas now makes them rank differently relative to 2015, as well as adding Extended Data Figure 7. The marginals in the new confusion graph are missing an explanation where they are derived from. I assume the horizontal row is the number of indications that are included in the current analysis, but had been missing in 2015, and the vertical column is the inverse? If so (or other), please add explanation to the Legend or graph.

Revised the ED7G legend: "Confusion matrix showing the categorization of unique drug indications into therapy areas in Nelson et al 2015 versus current. Note that the current categorization is based on each indication's position in the MeSH ontological tree and one indication can appear in >1 area, see Methods for details. *Marginals along the top edge are the number of drug indications in each current therapy area that were absent from the 2015 dataset. Marginals along the right edge are the number of drug indications in each 2015 therapy area that are absent from the current dataset.*"

The sentence introduced in response to my Comment 4 "When these GWAS quadrupled the number of T2D associated hit genes..." currently stands a bit isolated (and can probably be further simplified). It'd be good to refer or link these results to the sound new discussion on the diminishing returns topic further below in the manuscript.
Heiko Runz

Moved that sentence to the conclusion and added two linking sentences (changes italicized):
"We explored this question for T2D studies referenced above. *When these GWAS quadrupled*

the number of T2D associated genes from 217 to 862, new genetic support was identified for 7 of 95 mechanisms in clinical development while the number supported increased from 5 to 7 out of 12 launched drug mechanisms. Thus, RS has remained high in light of new GWAS data. One can also, however, consider the proportion of genetic associations that are successful drug targets. Of the 7 targets of launched drugs with genetic evidence, 4 had Mendelian evidence..."

Referee #2 (Remarks on code availability):

No, but I confirm it is accessible together with data under the provided link

Referee #3 (Remarks to the Author):

The authors have clearly and thoroughly addressed the comments I provided (and most of those from the other reviewers) and I believe the manuscript is improved as a result. In particular, the addition of ST1 and the definitions/abbreviations in the Supp Tables and main text should make the manuscript easier for readers to interpret and recreate results. While there are some analyses that would have been ideal to be able to incorporate, I understand that the available data is often limited and hence the authors have appropriately listed these as limitations in the text instead.

We thank Reviewer 3 for their helpful feedback and contributions.